# First-in-human phase I dose escalation trial of the first-in-class tumor microenvironment modulator VT1021 in advanced solid tumors

Devalingam Mahalingam [1✉], Wael Harb[2], Amita Patnaik[3], Andrea Bullock [4], Randolph S. Watnick[5], Melanie Y. Vincent[6], Jian Jenny Chen [6], Suming Wang[6], Harold Pestana[6], Judy Chao[6], James Mahoney[6], Michael Cieslewicz[6] & Jing Watnick [6✉]

## Abstract

**Background** VT1021 is a cyclic peptide that induces the expression of thrombospondin-1 (TSP-1) in myeloid-derived suppressor cells (MDSCs) recruited to the tumor microenvironment (TME). TSP-1 reprograms the TME via binding to CD36 and CD47 to induce tumor and endothelial cell apoptosis as well as immune modulation in the TME.

**Methods** Study VT1021-01 (ClinicalTrials.gov ID NCT03364400) used a modified $3 + 3$ design. The primary objective was to determine the recommended Phase 2 dose (RP2D) in patients with advanced solid tumors. Safety, tolerability, and pharmacokinetics (PK) were assessed. Patients were dosed twice weekly intravenously in 9 cohorts (0.5–15.6 mg/kg). Safety was evaluated using CTCAE version 5.0 and the anti-tumor activity was evaluated by RECIST version 1.1.

**Results** The RP2D of VT1021 is established at 11.8 mg/kg. VT1021 is well tolerated with no dose-limiting toxicities reported (0/38). The most frequent drug-related adverse events are fatigue (15.8%), nausea (10.5%), and infusion-related reactions (10.5%). Exposure increases proportionally from 0.5 to 8.8 mg/kg. The disease control rate (DCR) is 42.9% with 12 of 28 patients deriving clinical benefit including a partial response (PR) in one thymoma patient (504 days).

**Conclusions** VT1021 is safe and well-tolerated across all doses tested. RP2D has been selected for future clinical studies. PR and SD with tumor shrinkage are observed in multiple patients underscoring the single-agent potential of VT1021. Expansion studies in GBM, pancreatic cancer and other solid tumors at the RP2D have been completed and results will be communicated in a separate report.

## Plain language summary

It may be possible to treat cancers with therapies that modify the tumor microenvironment. This is the environment in the body in which tumors survive and grow and is composed of different types of cells. One such potential therapy is VT1021. Here, we conduct the first clinical trial to test this therapy in patients. We identify the optimal dose of the treatment to take into further studies, finding that VT1021 is safe and well tolerated by patients. We see some signs that the treatment is working in some patients and see evidence of modification of the tumor microenvironment. These findings help to inform further clinical trials of VT1021 to determine whether it is safe and effective in larger cohorts of patients.

[1] Northwestern University Medical School, Chicago, IL, USA. [2] Horizon Oncology Center, Lafayette, IN, USA. [3] South Texas Accelerated Research Therapeutics, San Antonio, TX, USA. [4] Beth Israel Deaconess Hospital, Boston, MA, USA. [5] Boston Children's Hospital, Boston, MA, USA. [6] Vigeo Therapeutics, Cambridge, MA, USA. ✉email: Mahalingam@nm.org; Jing.Watnick@vigeotx.com

Despite an increased understanding of the physiological processes involved in tumor metastasis, there are limited therapies that have proven clinical efficacy in advanced metastatic cancers such as glioblastoma, ovarian, prostate, pancreatic, and triple-negative breast cancer. The critical role of the TME as both a stimulator and a suppressor of tumor progression and metastasis is now widely recognized[1,2]. A potential TME-targeted therapy has been proposed where metastasis-incompetent tumors generate metastasis-suppressive microenvironments in distant organs by inducing TSP-1 expression in the bone marrow-derived Gr1+ myeloid cells[3,4].

A potent inhibitor of tumor metastasis, Prosaposin (Psap), acts via stimulation of p53 and the anti-tumorigenic TSP-1 in bone marrow-derived cells that are recruited to metastatic sites[5,6]. Within the TME, TSP-1 has been shown to act on two key receptors CD36 and CD47 (4,5). As a mediator of the pro-apoptotic activity of TSP-1, CD36 has been shown to be expressed on greater than 97% of human serous ovarian tumors tested (7). CD36 was also found to be increased in metastatic tumors compared to primary tumors, which are 2-3-fold higher than levels in ovarian and fallopian tube tissue[7]. CD36 has also been shown to be expressed in multiple human cancer cell lines including those derived from pancreatic, ovarian, breast, and prostate cancer[7,8]. Another TSP-1 receptor, CD47 is expressed in various types of cancer and has been shown to inhibit the direct killing of cancer cells[9] by binding to SIRPα on the cell surface of macrophages which represents a "do-not eat-me" signal to prevent phagocytosis by the macrophage[10]. CD47 also stimulates tumor-initiating cells, sometimes called cancer stem cells, to differentiate into mature cells[9]. High levels of either CD36 or CD47 are both prognostic indicators of poor outcomes for cancer patients[11,12]. Taken together, these findings suggest that a drug that stimulates expression of TSP-1 in the TME may have multiple beneficial effects as an anti-cancer agent.

To identify potential anti-cancer agents a proprietary TME screening platform was utilized to evaluate metastatic vs localized tumors and refractory vs responsive tumors. Based on these findings, VT1021 was developed with drug-like properties derived from the active sequence in Psap[4]. VT1021 exhibited TSP-1-inducing activity and significantly regressed tumors in a PDX model of metastatic ovarian cancer[7]. The in vivo activity of VT1021 in murine xenograft models with several human solid tumor indications is presented in this report. This first-in-human phase 1 study was designed to determine RP2D, investigate the safety, pharmacokinetics (PK), and efficacy as well as confirm the mechanism of action of the novel, first-in-class, dual inhibitor of CD36 and CD47, VT1021, in patients with advanced solid tumors. Here, we select RP2D for VT1021, demonstrate that it is safe and well tolerated at all dosing levels, achieving a disease control rate (DCR) of 42.9% with clear validation of the proposed mechanism of action via the stimulation of TSP-1 in the TME.

## Methods

**Clinical study design.** NCT03364400 was a phase 1, first-in-human, multicenter, open-label, dose escalation, and expansion study of VT1021 designed and sponsored by Vigeo Therapeutics, Inc. Data for the dose escalation phase are reported here. The data cut-off date was December 8, 2021. For the dose escalation portion of the study, the first patient was enrolled on November 28, 2017, and the last patient was enrolled on January 27, 2020.

The primary objective of the escalation phase was to determine the RP2D for VT1021. The secondary objectives were to characterize the adverse event (AE) profile, determine the PK, and describe preliminary evidence of efficacy, if feasible, by using objective response rate (ORR), disease control rate (DCR), and progression-free survival (PFS) based on Response Evaluation Criteria in Solid Tumors (RECIST) v1.1. Exploratory objectives included pharmacodynamic (PD) assessment of expression levels of CD36, CD47, TSP-1 and selected immune cells by immunohistochemistry (IHC) on pairs of pre- and on-study biopsies.

Eligible patients had advanced solid tumors that were refractory to, or intolerant of, existing therapies known to provide clinical benefit for their condition. Patients were aged ≥18 years and had Eastern Cooperative Oncology Group (ECOG) performance status of ≤2. Patients had evaluable or measurable disease by RECIST v1.1. Patients had to have adequate marrow reserve, liver and renal function. Key exclusion criteria included diagnosis of another malignancy within the past 2 years, history of a major surgical procedure or a significant traumatic injury within 14 days prior to commencing study drug, treatment with investigational therapy(ies) within 5 half-lives of the investigational therapy prior to the first scheduled day of dosing with VT1021, evidence of symptomatic brain metastases and use of other concurrent chemotherapy, immunotherapy, radiotherapy, or investigational anti-cancer therapy. Full eligibility criteria are available in the Protocol (Supplementary Information).

Dose escalation was a variation to the traditional 3 + 3 study design. The dose escalation consisted of the administration of VT1021 intravenously twice weekly at doses of 0.5, 1.0, 2.0, 3.3, 5.1, 6.6, 8.8, 11.8 or 15.6 mg/kg (Fig. 1). The starting dose was 1 mg/kg, established based on pre-clinical toxicity studies. Safety

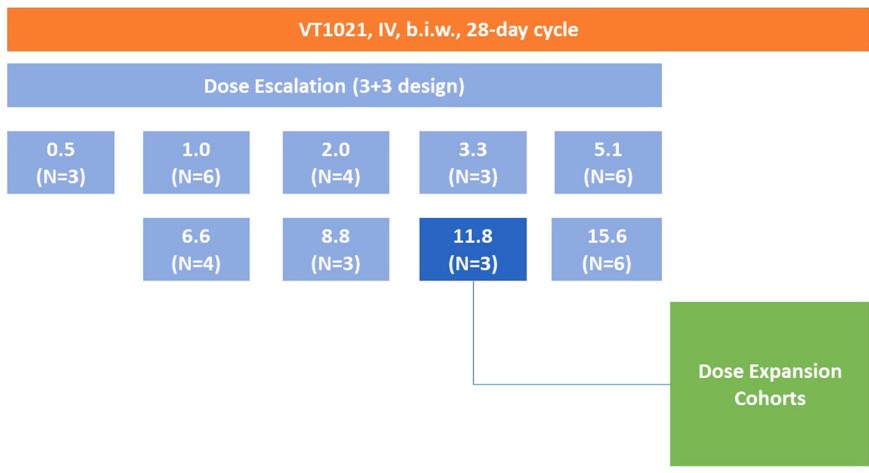

**Fig. 1** Study schema and participants of the phase 1 clinical trial of VT1021.

was evaluated using CTCAE version 5.0. Each dose level would enroll at least one patient. If no dose-limiting toxicity (DLT) was observed, the next patient would be enrolled at the next higher dose level. If one DLT was observed, a minimum of 3 patients must be treated at the same dose level. Dose escalation was to continue until at least 2 patients in a cohort of 6 experienced DLT. Patients received VT1021 by intravenous infusion twice weekly IV on a 28-day cycle. The extent of disease was evaluated by imaging studies at the end of Cycle 2 and after every 2 cycles thereafter. Treatment would continue until disease progression, unacceptable toxicity or another withdrawal criteria was met. Intra-patient dose escalation was permitted upon meeting pre-specified criteria. RP2D was defined as the dose level where ≤33% patients experienced DLT. DLTs (defined in the study protocol, Supplementary Information) were assessed during the first 28 days of treatment.

To decrease the risk of infusion reactions during the first week of dosing a premedication regimen was implemented. Prior to receiving each infusion of VT1021 patients were required to receive premedication with either dexamethasone by mouth 6–12 h pre-infusion or methylprednisolone 0.5 to 2 h prior to start of infusion and antihistamine (H1 antagonist), acetaminophen, and H2 blockers at the discretion of the investigator. In lieu of the premedication regimen clinical investigators were allowed to administer premedication regimes as per institutional guidelines. The premedication corticosteroid dose was to be decreased, tapered, or eliminated at the Investigator's discretion after the first week of dosing.

Patient assessment and follow-up procedures can be found in the schedule of assessments in the study protocol Appendix 1 (Supplementary Information). Per protocol, regular safety assessments were performed in a population of patients who have received at least one dose of VT1021, including but not limited to physical examinations, ECOG/Karnofsky PS, electrocardiograms, and laboratory parameters. Clinical response was evaluated by using RECIST v1.1 and iRECIST in a population of patients who have at least completed one cycle of VT1021 treatment, per protocol. Blood samples were collected for PK analysis on Days 1, 4, 8, 11, 15, 18, 22, 25 and 50 at pre-dose, and at 0, 2-, 4-, 6- and 24-h post-infusion (Fig. 2). Plasma concentrations were determined with a validated assay using liquid chromatography- mass spectrometry.

For patients who have signed a consent form, a pre-study biopsy or archival tumor specimen obtained within 6 months prior to study initiation was collected. In addition, on-study biopsies were collected at the end of Cycle 1 Week 4 or at any time during Cycle 2. Biopsies could be obtained after cycle 2 at the discretion of the Investigator. Paired pre-study and on-study biopsies were analyzed for expression of CD36, CD47, TSP-1, and immune cell populations by both IHC (Figs. 3–5) and MIBI (Multiplexed Ion Beam Imaging). MIBI is performed by staining formalin-fixed paraffin-embedded (FFPE) tissue with a panel of metal-labeled antibodies and then imaging the tissue using time-of-flight secondary ion mass spectrometry (ToF-SIMS)[13]. The masses of detected species are then assigned to target biomolecules given the unique metal isotope label of each antibody, creating multiplexed images. All antibodies in the panel have been MIBI validated on human FFPE tissue.

All relevant ethical regulations were followed during the study. The methods were performed in accordance with relevant guidelines and regulations and approved by the Food and Drug Administration (FDA). Written informed consent was obtained from all patients who participated in the study. The Institutional Review Boards (IRBs) in all participating institutions have approved the study protocol. The institutions participated in the study are Northwestern University Medical School, Chicago, IL, Horizon Oncology Center, Lafayette, IN, South Texas Accelerated Research Therapeutics, San Antonio, TX, and Beth Israel Deaconess Hospital, Boston, MA.

We used the CONSORT checklist when writing our report[14].

**Inclusion criteria.** To qualify for enrollment, all the following criteria must be met: (1) Patient must provide written informed consent. (2) Patient is ≥18 years of age. (3) For the Dose Escalation Phase: Patients with advanced solid tumors that are refractory to, or intolerant of, existing therapies known to provide clinical benefit for their condition. (4) Patient has evaluable or measurable disease by RECIST v1.1. (5) Patient has a performance status (PS) of 0–1 on the Eastern Cooperative Oncology Group (ECOG) scale. (6) Patient is at least 21 days removed from therapeutic radiation or chemotherapy prior to the first scheduled day of dosing with VT1021 and has recovered to Grade ≤ 1 (National Cancer Institute [NCI] Common Terminology Criteria for Adverse Events [CTCAE] v5.0) from all clinically significant toxicities related to prior therapies. (a) For patients receiving nitrosoureas or mitomycin C, the window is 6 weeks. (b) For patients receiving monoclonal antibody therapy, the window is at least one half-life or 4 weeks (whichever is shorter). (7) Patient has adequate organ function defined as: (a) Absolute neutrophil count (ANC) $\geq 1.5 \times 10^9$/L (1500/μL) and absolute lymphocyte count (ALC) $\geq 7 \times 10^9$/L (700/μL). (b) Platelet $\geq 100 \times 10^9$/L. (c) Hemoglobin ≥9 g/dL. (d) Activated partial thromboplastin time/ prothrombin time/international normalized ratio (aPTT/PT/ INR) ≤ 1.5 × upper limit of normal (ULN) unless the patient is on anticoagulants in which case therapeutically acceptable values (as determined by the investigator) meet eligibility requirements. (e) Aspartate aminotransferase (AST) or alanine aminotransferase (ALT) ≤ 2.5 × ULN. In the case of known (i.e., radiological or biopsy documented) liver metastasis, serum transaminase levels must be ≤5 × ULN. (f) Total serum bilirubin ≤1.5 × ULN (except for patients with known Gilbert's Syndrome ≤3 × ULN is permitted). (g) Renal: Serum creatinine <2.0 × ULN and creatinine clearance ≥50 L/min/1.73 m². (h) Serum albumin >3 gm/dL. (8) Patient agrees to use acceptable methods of contraception during the study and for at least 90 days after the last dose of VT1021 if sexually active and able to bear or beget children.

**Exclusion criteria.** The presence of any of the following will exclude the patient from the study: (1) Diagnosis of another malignancy within the past 2 years (excluding a history of carcinoma in situ of the cervix, superficial non-melanoma skin cancer, or superficial bladder cancer that has been adequately treated, or stage 1 prostate cancer that does not require treatment or requires only treatment with luteinizing hormone-releasing hormone agonists or antagonists if initiated at least 90 days prior to the first dose of VT1021). (2) History of a major surgical procedure or a significant traumatic injury within 14 days prior to commencing study drug, or the anticipation of the need for a major surgical procedure during the course of the study. (3) Treatment with investigational therapy(ies) within 5 half-lives of the investigational therapy prior to the first scheduled day of dosing with VT1021, or 4 weeks if the half-life of the investigational agent is not known, whichever is shorter. (4) Concurrent serious (as determined by the Principal Investigator [PI]) medical conditions, including, but not limited to, New York Heart Association (NYHA) class III or IV congestive heart failure, history of congenital prolonged QT syndrome, uncontrolled infection, active hepatitis B, hepatitis C or human immunodeficiency virus (HIV), or other significant co-morbid conditions that, in the opinion of the Investigator, would impair study participation or cooperation. (5) Pregnant or planning to become

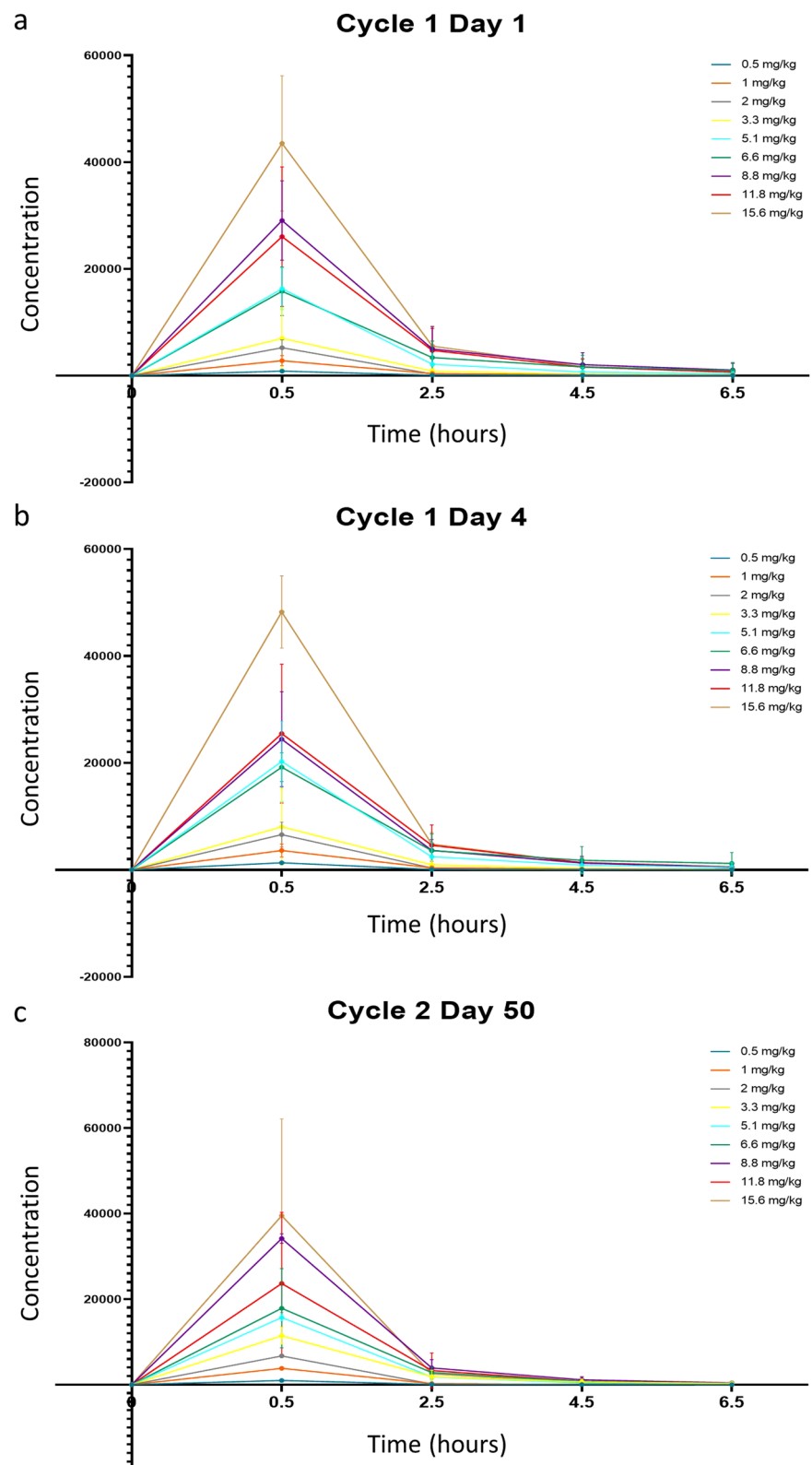

**Fig. 2 Concentration-time profiles for VT1021, by dose cohort. a** Cycle 1 day 1. **b** Cycle 1 day 4. **c** Cycle 2 day 50.

pregnant or breast feed while on study. (6) Evidence of symptomatic brain metastases. Patients with treated (surgically excised or irradiated) and stable brain metastases are eligible, assuming the patient has adequately recovered from treatment, the treatment was at least 28 days prior to initiation of study drug, and baseline brain computed tomography (CT) with contrast or

magnetic resonance imaging (MRI) within 14 days of initiation of study drug, is negative for new or worsening brain metastases. (7) Other concurrent chemotherapy, immunotherapy, radiotherapy, or investigational anti-cancer therapy. (8) Requirement for palliative radiotherapy to lesions that are defined as target lesions by RECIST/RANO criteria at the time-of study entry. (9) Known

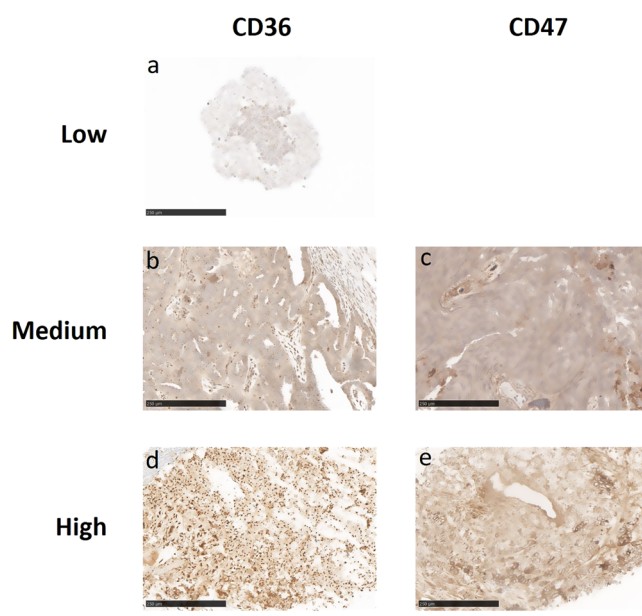

**Fig. 3 Representative images of CD36 and CD47 expression intensity by Immunohistochemistry. a** Low CD36 expression from a patient with pancreatic cancer, **b** Medium CD36 expression from a patient with ovarian cancer. **c** Medium CD47 expression from a patient with non-small cell lung cancer. **d** High expression of CD36 and **e** high expression of CD47 from a patient with uterine cancer. No low CD47 expression was observed in any of the biopsies. Bar denotes 250 µm.

hypersensitivity to any of the components of VT1021 (sodium phosphate dibasic anhydrous, sodium phosphate monobasic monohydrate, mannitol, polysorbate 80) or a severe reaction to PS20- or PS80-containing drugs or investigational agents (e.g. amiodarone, Vitamin K, etoposide, docetaxel, cancer vaccine, protein biotherapeutics [like monoclonal antibodies], erythropoietin-stimulating agents, fosaprepitant). (10) Chronic, systemically administered glucocorticoids in doses equivalent to >5 mg prednisone daily. Topical, inhalational, ophthalmic, intraarticular, and intranasal glucocorticoids are permitted. Isolated or intermittent use of systemically administered glucocorticoids to treat complications of malignancy, use as a premedication, or as a onetime prep for an imaging procedure is permitted. If patient was on >5 mg prednisone/day equivalent, last dose must have been at least 7 days prior to the first planned dose of study drug. (11) Patients with active hepatitis B (e.g., hepatitis B surface antigen [HBsAg] reactive) are excluded, however, patients with past hepatitis B virus (HBV) infection or resolved HBV infection (defined as the presence of hepatitis B core antibody [HBcAb] and absence of HBsAg) may be enrolled provided that prior testing/known status for HBV deoxyribonucleic acid (DNA) is negative. Patients with active hepatitis C (e.g., hepatitis C virus [HCV] ribonucleic acid [RNA] [qualitative] are detected) are excluded, however, patients with cured hepatitis C (negative HCV RNA prior test/known status) may be enrolled.

**Statistical analysis.** The disease control rate (DCR) used for clinical outcomes was calculated as the percentage of patients with advanced cancer whose therapeutic intervention has led to a

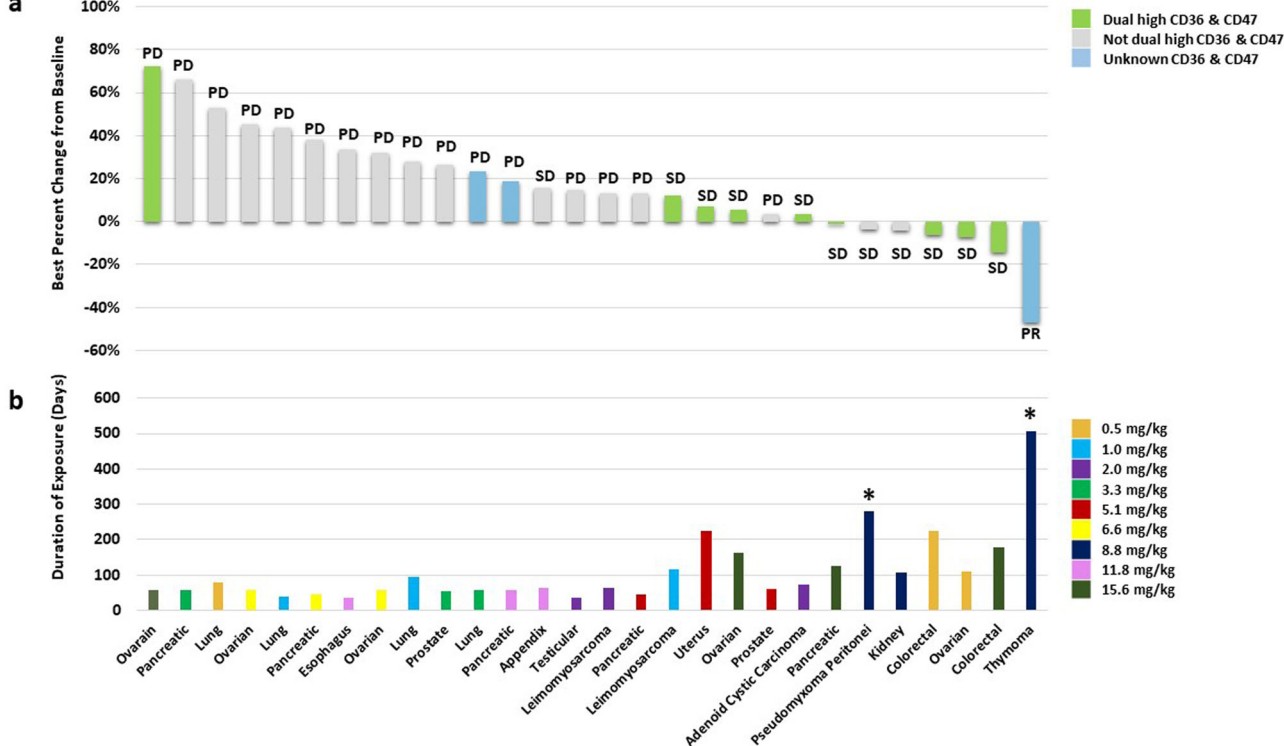

**Fig. 4 Percentage change from baseline in target lesions and duration of treatment with VT1021. a** Percentage change from baseline was determined for target tumor lesions obtained at best response calculated as the percentage change from baseline. The best overall response is shown for each patient according to RECIST v1.1. Expression intensities of CD36 and CD47 are indicated as dual high (green), not dual high (gray), or unknown (blue). **b** Duration of exposure to VT1021 was determined as the period from cycle 1 day date to the end of treatment date. Dose cohorts are indicated by color. *Patients dose-escalated from 8.8 to 11.8 mg/kg. PR partial response, SD stable disease, PD progressive disease.

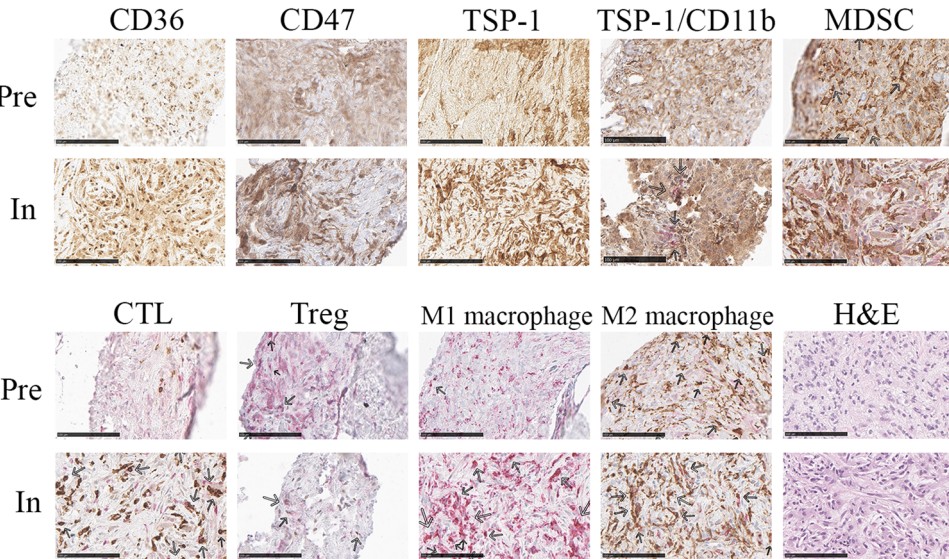

**Fig. 5 VT1021 Reprograms the tumor microenvironment by inducing Thrombospodin-1 (TSP-1) expression.** Immunohistochemistry was performed on pre- and on-study tumor biopsies (both from the liver) of a patient with metastatic Renal Cell Carcinoma, dosed at 8.8 mg/kg VT1021. Upper Panel: CD36, CD47, TSP-1, TSP-1/CD11b, and myeloid-derived suppressor cells (CD11bCD14). Lower Panel: Cytotoxic T cells (CD3 + /CD8 + ), Tregs (CD3 + / FoxP3 + ), M1 macrophages (CD68 + /iNOS + ) and M2 macrophages (CD68 + /CD163 + ). Bar denotes 100 μm. iNOS inducible nitric oxide synthase.

complete response, partial response, or stable disease. The 90% Confidence Interval (CI) was calculated using the exact (Clopper-Pearson) interval.

Statistical analysis was performed with Graphpad Prism 9.3.1, $p$ values were calculated by unpaired two-sample $t$-test, graphs of a point with error bars are used to indicate the average values and standard error of the mean (SEM).

**Reporting summary**. Further information on research design is available in the Nature Portfolio Reporting Summary linked to this article.

## Results

The RP2D for VT1021 was determined to be 11.8 mg/kg, twice weekly dosing. The consideration was based on the combination of assessment of safety, tolerability, as well as PK exposure across all dose levels. The individual assessments of these parameters are described in the following sections.

**Clinical patient population, treatment, and disposition**. Thirty-eight patients, who received at least 1 dose of VT1021, were enrolled (Fig. 1). Patient demographics and disease characteristics by dose cohort are shown in Table 1. The median age was 65 years (range 40–84) and 100% of patients had an ECOG performance status of ≤2. A variety of tumor types enrolled most commonly being ovarian cancer (8 patients, 21%) and pancreatic cancer (7 patients, 19%); other tumor types had three or fewer patients (Table 1).

The patient population was heavily pre-treated; all patients received prior antineoplastic therapy with 78.9% of patients receiving ≥3 prior treatment regimens for advanced/metastatic disease, with 42% receiving prior radiotherapy.

Intra-patient dose escalation was permitted and a total of 4 patients were dose-escalated: one patient with colorectal cancer was dose-escalated from 0.5 to 1.0 mg/kg at cycle 8, one patient with uterine cancer was dose-escalated from 5.1 to 6.6 mg/kg at cycle 7, and two patients, one with thymoma and the other with pseudomyxoma peritonei, were dose-escalated from 8.8 to 11.8 mg/kg at cycles 7 and 8, respectively. One patient with

ovarian cancer dose de-escalated from 6.6 to 5.1 mg/kg at cycle 2 due to concern of worsening baseline peripheral neuropathy from grade 1 to 2, which was later attributed to previous platinum chemotherapy.

To achieve the minimum of 3 evaluable patients dose levels 1.0 mg/kg, 5.1 mg/kg and 15.6 mg/kg were over-enrolled. Some of the patients withdrew voluntarily, others withdrew to go to hospice care, at least 1 patient withdrew due to an infusion reaction during the first dose.

The first patient dosed at the protocol-defined starting dose of 1 mg/kg experienced a grade 3 infusion reaction on the third dose. To ensure safety, 3 patients were treated at the lower dose level of 0.5 mg/kg and the protocol was amended to require premedication of steroids and/or antihistamines (at the PI's discretion) prior to the first infusion. No infusion reaction was reported in 3 patients treated at 0.5 mg/kg and dose escalation resumed. Every patient received premedication per protocol and the majority of the patients were tapered off the premedication after 1–2 weeks of VT1021 treatment.

Of the patients that were discontinued from the study, reasons for discontinuation included disease progression (65.8%), patient or physician decision (13.2%), AEs (10.5%) and death (7.9%). None of the deaths were attributed to VT1021 treatment. Four patients died on treatment, and the cause of death were hepatic failure due to disease progression (at 5.1 mg/kg dose level), hepatic failure due to disease progression (at 6.6 mg/kg dose level), tumor hemorrhage (at 15.6 mg/kg dose level) and septic shock/unrelated to protocol (at 15.6 mg/kg dose level). Three patients died after treatment discontinuation within 30 days following the last dose of VT1021, and the cause of death were disease progression (at 3.3 mg/kg dose level), multi-organ failure (at 11.8 mg/kg dose level), and disease progression (at 15.6 mg/kg dose level).

**Safety and tolerability**. Overall, VT1021 had a clean safety profile. The incidence of ≥grade 3 AEs suspected to be related to the study drug was very low (7.9%). Thirty-seven patients (97.4%) experienced at least one treatment-emergent adverse event (TEAE) and 17 patients (44.7%) experienced grade ≥3 TEAEs

**Table 1 Baseline patient demographics and characteristics, by treatment group.**

| | 0.5 mg/kg n = 3 | 1.0 mg/kg n = 6 | 2.0 mg/kg n = 4 | 3.3 mg/kg n = 3 | 5.1 mg/kg n = 6 | 6.6 mg/kg n = 4 | 8.8 mg/kg n = 3 | 11.8 mg/kg n = 3 | 15.6 mg/kg n = 6 | All patients n = 38 |
|---|---|---|---|---|---|---|---|---|---|---|
| Median age, years (range) | 69 (63–82) | 70.5 (46–73) | 46.5 (40–67) | 78 (67–83) | 65 (57–84) | 64.5 (60–83) | 64 (64–66) | 58 (54–61) | 65 (54–76) | 65 (40–84) |
| Sex (male), n (%) | 2 (66.7) | 4 (66.7) | 1 (25.0) | 1 (33.3) | 3 (50.0) | 0 | 1 (33.3) | 2 (66.7) | 2 (33.3) | 16 (42.1) |
| Race, n (%) | | | | | | | | | | |
| Caucasian | 3 (100) | 5 (83.3) | 2 (50.0) | 3 (100) | 4 (66.7) | 4 (100) | 3 (100) | 3 (100) | 5 (83.3) | 32 (84.2) |
| Black | 0 | 1 (16.7) | 2 (50.0) | 0 | 1 (16.7) | 0 | 0 | 0 | 1 (16.7) | 5 (13.2) |
| Asian | 0 | 0 | 0 | 0 | 1 (16.7) | 0 | 0 | 0 | 0 | 1 (2.6) |
| ECOG PS, n (%) | | | | | | | | | | |
| 0 | 0 | 0 | 2 (50.0) | 1 (33.3) | 3 (50.0) | 2 (50.0) | 0 | 1 (33.3) | 0 | 9 (23.7) |
| 1 | 3 (100) | 6 (100) | 1 (25.0) | 2 (66.7) | 3 (50.0) | 2 (50.0) | 3 (100) | 2 (66.7) | 6 (100) | 28 (73.7) |
| 2 | 0 | 0 | 1 (25.0) | 0 | 0 | 0 | 0 | 0 | 0 | 1 (2.6) |
| Primary tumor type, n (%) | | | | | | | | | | (n = 38) |
| Adenoid cystic carcinoma | 0 | 0 | 1 (25.0) | 0 | 0 | 0 | 0 | 0 | 0 | 1 (2.6) |
| Anal cancer | 0 | 0 | 0 | 0 | 0 | 0 | 0 | 0 | 1 (16.7) | 1 (2.6) |
| Appendiceal cancer | 0 | 0 | 0 | 0 | 0 | 0 | 0 | 1 (33.3) | 0 | 1 (2.6) |
| Uterine carcinosarcoma | 0 | 0 | 0 | 0 | 1 (16.7) | 0 | 0 | 0 | 0 | 1 (2.6) |
| Cholangiocarcinoma | 0 | 0 | 1 (25.0) | 0 | 0 | 0 | 0 | 0 | 0 | 1 (2.6) |
| Colorectal cancer | 1 (33.3) | 0 | 0 | 0 | 1 (16.7) | 0 | 0 | 0 | 1 (16.7) | 3 (7.9) |
| Esophageal cancer | 0 | 0 | 0 | 0 | 1 (16.7) | 0 | 0 | 1 (33.3) | 0 | 2 (5.3) |
| Head and neck cancer | 0 | 1 (16.7) | 0 | 0 | 0 | 0 | 0 | 0 | 0 | 1 (2.6) |
| Leiomyosarcoma | 0 | 1 (16.7) | 1 (25.0) | 0 | 0 | 0 | 0 | 0 | 0 | 2 (5.3) |
| Non-small cell lung cancer | 1 (33.3) | 1 (16.7) | 0 | 1 (33.3) | 0 | 0 | 0 | 0 | 0 | 3 (7.9) |
| Ovarian cancer | 1 (33.3) | 0 | 0 | 0 | 1 (16.7) | 3 (75.0) | 0 | 0 | 3 (50.0) | 8 (21.1) |
| Pancreatic cancer | 0 | 2 (33.3) | 0 | 1 (33.3) | 1 (16.7) | 1 (25.0) | 0 | 1 (33.3) | 1 (16.7) | 7 (18.4) |
| Prostate cancer | 0 | 0 | 0 | 1 (33.3) | 1 (16.7) | 0 | 0 | 0 | 0 | 2 (5.3) |
| Pseudomyxoma peritonei | 0 | 0 | 0 | 0 | 0 | 0 | 1 (33.3) | 0 | 0 | 1 (2.6) |
| Renal cell carcinoma | 0 | 0 | 0 | 0 | 0 | 0 | 1 (33.3) | 0 | 0 | 1 (2.6) |
| Small cell lung cancer | 0 | 1 (16.7) | 0 | 0 | 0 | 0 | 0 | 0 | 0 | 1 (2.6) |
| Testicular cancer | 0 | 0 | 1 (25.0) | 0 | 0 | 0 | 0 | 0 | 0 | 1 (2.6) |
| Thymoma | 0 | 0 | 0 | 0 | 0 | 0 | 1 (33.3) | 0 | 0 | 1 (2.6) |
| Prior treatment regimens, n (%) | | | | | | | | | | |
| 1 | 1 (33.3) | 1 (16.7) | 0 | 0 | 0 | 0 | 2 (66.7) | 0 | 0 | 4 (10.5) |
| 2 | 0 | 0 | 0 | 1 (33.3) | 1 (16.7) | 0 | 0 | 1 (33.3) | 1 (16.7) | 4 (10.5) |
| ≥3 | 2 (66.7) | 5 (83.3) | 4 (100.0) | 2 (66.7) | 5 (83.3) | 4 (100.0) | 1 (33.3) | 2 (66.7) | 5 (83.3) | 30 (78.9) |
| Prior Radiotherapy, n (%) | | | | | | | | | | |
| Yes | 1 (33.3) | 5 (83.3) | 2 (50.0) | 1 (33.3) | 0 | 1 (25.0) | 1 (33.3) | 2 (66.7) | 3 (50.0) | 16 (42.1) |
| No | 2 (66.7) | 1 (16.7) | 2 (50.0) | 2 (66.7) | 6 (100) | 3 (75.0) | 2 (66.7) | 1 (33.3) | 3 (50.0) | 22 (57.9) |

ECOG PS Eastern Cooperative Oncology Group performance status.

shown in Table 2 (TEAEs in ≥5% of patients). A TEAE is defined as any event that occurs on or after the first dose of study drug administration or any pre-existing event which worsened in severity after dosing. There were 5 patients (13.2%) with fatal TEAEs, none of which were classified as drug-related. AEs suspected to be related to the study treatment (RTEAEs) were experienced by 18 patients (47.4%) shown in Table 3 (RTEAEs in ≥5% of patients); the most frequent RTEAEs (≥10% of patients) were fatigue (6 patients, 15.8%), nausea (4 patients, 10.5%) and infusion-related reaction (4 patients, 10.5%). Grade 3 RTEAEs were reported in 3 patients (7.9%) where there was a single occurrence each of infusion-related reaction, anemia, and increased aspartate aminotransferase (AST), blood bilirubin and creatinine. Study drug was held for grade 3 elevation in AST and blood bilirubin and the patient was discontinued from treatment for clinical progression of disease. Study drug was discontinued for the patient with grade 3 infusion reaction and not re-started

for the patient with anemia and increased blood creatinine who subsequently experienced an SAE of sepsis.

**DLTs, PK and RP2D.** Throughout the course of the dose escalation trial no patient experienced a DLT and thus MTD was not achieved. Because no MTD was reached, the recommended phase 2 dose (RP2D) was determined based on the pharmacokinetic (PK) profile. Table 4 shows the PK parameters for VT1021 by dose cohort and Fig. 2 shows the median concentration-time profiles by dose. VT1021 plasma exposures increased dose proportionally from 0.5 to 8.8 mg/kg based on mean $C_{max}$, AUC, and CL values and the exposures from 8.8 mg/kg to 15.6 mg/kg were similar. VT1021 did not appear to accumulate in plasma with repeated dosing, which is consistent with the dosing frequency and short terminal half-life values observed (average 1.2 to 1.3 h across all doses and sampling days).

**Table 2 Treatment-emergent adverse events.**

| Preferred term, n (%) | 0.5 mg/kg n = 3 All | Gr ≥ 3 | 1.0 mg/kg n = 6 All | Gr ≥ 3 | 2.0 mg/kg n = 4 All | Gr ≥ 3 | 3.3 mg/kg n = 3 All | Gr ≥ 3 | 5.1 mg/kg n = 6 All | Gr ≥ 3 | 6.6 mg/kg n = 4 All | Gr ≥ 3 | 8.8 mg/kg n = 3 All | Gr ≥ 3 | 11.8 mg/kg n = 3 All | Gr ≥ 3 | 15.6 mg/kg n = 6 All | Gr ≥ 3 | All patients n = 38 All | Gr ≥ 3 |
|---|---|---|---|---|---|---|---|---|---|---|---|---|---|---|---|---|---|---|---|---|
| Total | 3 (100) | 0 | 6 (100) | 2 (33.3) | 3 (75.0) | 2 (50.0) | 3 (100) | 1 (33.3) | 6 (100) | 5 (83.3) | 4 (100) | 1 (25.0) | 3 (100) | 1 (33.3) | 3 (100) | 1 (33.3) | 6 (100) | 4 (66.7) | 37 (97.4) | 17 (44.7) |
| Fatigue | 0 | 0 | 1 (16.7) | 0 | 1 (25.0) | 0 | 1 (33.3) | 0 | 1 (16.7) | 0 | 0 | 0 | 0 | 0 | 2 (66.7) | 1 (33.3) | 3 (50.0) | 0 | 8 (21.1) | 1 (2.6) |
| Abdominal pain | 0 | 0 | 2 (33.3) | 1 (16.7) | 1 (25.0) | 0 | 0 | 0 | 1 (16.7) | 0 | 1 (25.0) | 0 | 1 (33.3) | 0 | 2 (66.7) | 0 | 1 (16.7) | 0 | 7 (18.4) | 2 (5.3) |
| Constipation | 0 | 0 | 1 (16.7) | 1 (16.7) | 0 | 0 | 0 | 0 | 1 (16.7) | 0 | 0 | 0 | 2 (66.7) | 0 | 2 (66.7) | 0 | 1 (16.7) | 0 | 7 (18.4) | 1 (2.6) |
| Nausea | 1 (33.3) | 0 | 0 | 0 | 0 | 0 | 0 | 0 | 0 | 0 | 1 (25.0) | 0 | 2 (66.7) | 0 | 1 (33.3) | 0 | 2 (33.3) | 0 | 6 (15.8) | 0 |
| Anemia | 0 | 0 | 1 (16.7) | 1 (16.7) | 1 (25.0) | 0 | 1 (33.3) | 0 | 0 | 0 | 0 | 0 | 1 (33.3) | 0 | 1 (33.3) | 1 (33.3) | 1 (16.7) | 1 (16.7) | 5 (13.2) | 3 (7.9) |
| Urinary tract infection | 0 | 0 | 1 (16.7) | 0 | 1 (25.0) | 0 | 1 (33.3) | 0 | 1 (16.7) | 0 | 0 | 0 | 0 | 0 | 0 | 0 | 1 (16.7) | 0 | 5 (13.2) | 0 |
| Arthralgia | 1 (33.3) | 0 | 0 | 0 | 0 | 0 | 0 | 0 | 1 (16.7) | 0 | 2 (50.0) | 0 | 1 (33.3) | 0 | 0 | 0 | 0 | 0 | 5 (13.2) | 0 |
| Dyspnoea | 1 (33.3) | 0 | 0 | 0 | 1 (25.0) | 0 | 0 | 0 | 0 | 0 | 2 (50.0) | 0 | 1 (33.3) | 0 | 0 | 0 | 0 | 0 | 5 (13.2) | 1 (2.6) |
| Infusion-related reaction | 0 | 0 | 2 (33.3) | 1 (16.7) | 0 | 0 | 0 | 0 | 0 | 0 | 1 (25.0) | 0 | 1 (33.3) | 0 | 0 | 0 | 0 | 0 | 4 (10.5) | 1 (2.6) |
| Blood bilirubin increased | 0 | 0 | 1 (16.7) | 0 | 0 | 0 | 0 | 0 | 0 | 0 | 1 (25.0) | 1 (25.0) | 0 | 0 | 1 (33.3) | 1 (33.3) | 1 (16.7) | 0 | 4 (10.5) | 2 (5.3) |
| Decreased appetite | 0 | 0 | 0 | 0 | 0 | 0 | 0 | 0 | 0 | 0 | 1 (25.0) | 0 | 1 (33.3) | 0 | 1 (33.3) | 0 | 1 (16.7) | 0 | 4 (10.5) | 0 |
| Hyperuricaemia | 1 (33.3) | 0 | 0 | 0 | 0 | 0 | 0 | 0 | 1 (16.7) | 0 | 1 (25.0) | 0 | 1 (33.3) | 0 | 0 | 0 | 0 | 0 | 4 (10.5) | 0 |
| Hypokalaemia | 0 | 0 | 1 (16.7) | 0 | 0 | 0 | 0 | 0 | 1 (16.7) | 0 | 1 (25.0) | 0 | 0 | 0 | 0 | 0 | 1 (16.7) | 0 | 4 (10.5) | 0 |
| Hypomagnesaemia | 1 (33.3) | 0 | 0 | 0 | 0 | 0 | 0 | 0 | 2 (33.3) | 0 | 0 | 0 | 0 | 0 | 0 | 0 | 1 (16.7) | 0 | 4 (10.5) | 0 |
| Headache | 1 (33.3) | 0 | 0 | 0 | 0 | 0 | 1 (33.3) | 0 | 0 | 0 | 0 | 0 | 1 (33.3) | 0 | 0 | 0 | 1 (16.7) | 0 | 4 (10.5) | 0 |
| Abdominal distension | 0 | 0 | 0 | 0 | 0 | 0 | 0 | 0 | 0 | 0 | 0 | 0 | 1 (33.3) | 1 (33.3) | 0 | 0 | 2 (33.3) | 1 (16.7) | 3 (7.9) | 2 (5.3) |
| Vomiting | 0 | 0 | 1 (16.7) | 0 | 0 | 0 | 0 | 0 | 0 | 0 | 0 | 0 | 1 (33.3) | 0 | 0 | 0 | 2 (33.3) | 0 | 3 (7.9) | 0 |
| Oedema peripheral | 0 | 0 | 1 (16.7) | 0 | 0 | 0 | 0 | 0 | 0 | 0 | 0 | 0 | 1 (33.3) | 0 | 1 (33.3) | 0 | 0 | 0 | 3 (7.9) | 0 |
| Oral candidiasis | 0 | 0 | 1 (16.7) | 0 | 0 | 0 | 0 | 0 | 0 | 0 | 1 (25.0) | 0 | 1 (33.3) | 0 | 0 | 0 | 0 | 0 | 3 (7.9) | 0 |
| Aspartate aminotransferase increased | 0 | 0 | 1 (16.7) | 0 | 0 | 0 | 0 | 0 | 1 (16.7) | 0 | 1 (25.0) | 1 (25.0) | 0 | 0 | 0 | 0 | 0 | 0 | 3 (7.9) | 1 (2.6) |
| Back pain | 0 | 0 | 0 | 0 | 0 | 0 | 0 | 0 | 1 (16.7) | 0 | 0 | 0 | 0 | 0 | 0 | 0 | 2 (33.3) | 0 | 3 (7.9) | 0 |
| Diarrhea | 0 | 0 | 0 | 0 | 0 | 0 | 0 | 0 | 1 (16.7) | 0 | 0 | 0 | 0 | 0 | 1 (33.3) | 0 | 0 | 0 | 2 (5.3) | 0 |
| Chills | 0 | 0 | 0 | 0 | 1 (25.0) | 0 | 0 | 0 | 0 | 0 | 0 | 0 | 0 | 0 | 0 | 0 | 1 (16.7) | 0 | 2 (5.3) | 0 |
| Pyrexia | 0 | 0 | 0 | 0 | 1 (25.0) | 0 | 0 | 0 | 1 (16.7) | 0 | 0 | 0 | 0 | 0 | 0 | 0 | 0 | 0 | 2 (5.3) | 0 |
| Hepatic failure | 0 | 0 | 0 | 0 | 0 | 0 | 0 | 0 | 1 (16.7) | 1 (16.7) | 1 (25.0) | 1 (25.0) | 0 | 0 | 0 | 0 | 0 | 0 | 2 (5.3) | 2 (5.3) |
| Sinusitis | 0 | 0 | 0 | 0 | 1 (25.0) | 0 | 0 | 0 | 1 (16.7) | 0 | 0 | 0 | 0 | 0 | 0 | 0 | 0 | 0 | 2 (5.3) | 0 |
| Alanine aminotransferase increased | 0 | 0 | 0 | 0 | 0 | 0 | 0 | 0 | 1 (16.7) | 0 | 1 (25.0) | 0 | 0 | 0 | 0 | 0 | 0 | 0 | 2 (5.3) | 0 |
| Blood alkaline phosphatase increased | 0 | 0 | 1 (16.7) | 0 | 0 | 0 | 0 | 0 | 1 (16.7) | 0 | 0 | 0 | 0 | 0 | 0 | 0 | 0 | 0 | 2 (5.3) | 0 |
| Blood creatine increased | 0 | 0 | 1 (16.7) | 0 | 0 | 0 | 0 | 0 | 0 | 0 | 1 (25.0) | 0 | 0 | 0 | 0 | 0 | 1 (16.7) | 1 (16.7) | 2 (5.3) | 1 (2.6) |
| White blood cell count decreased | 0 | 0 | 0 | 0 | 0 | 0 | 0 | 0 | 0 | 0 | 0 | 0 | 0 | 0 | 0 | 0 | 1 (16.7) | 0 | 2 (5.3) | 0 |
| Dizziness | 0 | 0 | 1 (16.7) | 0 | 0 | 0 | 0 | 0 | 0 | 0 | 0 | 0 | 0 | 0 | 0 | 0 | 1 (16.7) | 0 | 2 (5.3) | 0 |
| Peripheral sensory neuropathy | 0 | 0 | 0 | 0 | 0 | 0 | 0 | 0 | 0 | 0 | 0 | 0 | 0 | 0 | 1 (33.3) | 0 | 1 (16.7) | 0 | 2 (5.3) | 0 |
| Anxiety | 0 | 0 | 0 | 0 | 0 | 0 | 0 | 0 | 1 (16.7) | 0 | 0 | 0 | 0 | 0 | 1 (33.3) | 1 (33.3) | 0 | 0 | 2 (5.3) | 1 (2.6) |
| Vaginal hemorrhage | 0 | 0 | 0 | 0 | 1 (25.0) | 0 | 0 | 0 | 0 | 0 | 0 | 0 | 0 | 0 | 1 (33.3) | 0 | 0 | 0 | 2 (5.3) | 0 |
| Wheezing | 0 | 0 | 0 | 0 | 0 | 0 | 0 | 0 | 1 (16.7) | 0 | 0 | 0 | 0 | 0 | 1 (33.3) | 0 | 0 | 0 | 2 (5.3) | 0 |
| Rash maculo-papular | 1 (33.3) | 0 | 0 | 0 | 0 | 0 | 0 | 0 | 1 (16.7) | 0 | 0 | 0 | 0 | 0 | 0 | 0 | 0 | 0 | 2 (5.3) | 0 |
| Deep vein thrombosis | 0 | 0 | 0 | 0 | 0 | 0 | 1 (33.3) | 0 | 0 | 0 | 1 (25.0) | 0 | 0 | 0 | 0 | 0 | 0 | 0 | 2 (5.3) | 0 |

Treatment-Emergent Adverse Events is defined as any event that occurs on or after the first dose of study drug administration or any pre-existing event which worsened in severity after dosing. Safety Population - Dose Escalation = Patients with at least one dose. Dose groups represent Patients' initial dose. Patients with multiple unique events are counted once per each unique preferred term and System Organ Class. Coding used MedDRA version 23.0 Any event not graded will have a default value of grade 3.

COMMUNICATIONS MEDICINE | (2024)4:10 | https://doi.org/10.1038/s43856-024-00433-x | www.nature.com/commsmed

**Table 3 Related Treatment-emergent adverse events.**

| Preferred term, n (%) | 0.5 mg/kg n = 3 | | 1.0 mg/kg n = 6 | | 2.0 mg/kg n = 4 | | 3.3 mg/kg n = 3 | | 5.1 mg/kg n = 6 | | 6.6 mg/kg n = 4 | | 8.8 mg/kg n = 3 | | 11.8 mg/kg n = 3 | | 15.6 mg/kg n = 6 | | All Patients n = 38 | |
|---|---|---|---|---|---|---|---|---|---|---|---|---|---|---|---|---|---|---|---|---|
| | All | Gr ≥ 3 | All | Gr ≥ 3 | All | Gr ≥ 3 | All | Gr ≥ 3 | All | Gr ≥ 3 | All | Gr ≥ 3 | All | Gr ≥ 3 | All | Gr ≥ 3 | All | Gr ≥ 3 | All | Gr ≥ 3 |
| Total | 1 (33.3) | 0 | 3 (50.0) | 1 (16.7) | 1 (25.0) | 0 | 1 (33.3) | 0 | 2 (33.3) | 0 | 3 (75.0) | 1 (25.0) | 1 (33.3) | 0 | 3 (100) | 0 | 3 (50.0) | 1 (16.7) | 18 (47.4) | 3 (7.9) |
| Fatigue | 0 | 0 | 0 | 0 | 1 (25.0) | 0 | 1 (33.3) | 0 | 0 | 0 | 0 | 0 | 0 | 0 | 1 (33.3) | 0 | 3 (50.0) | 0 | 6 (15.8) | 0 |
| Nausea | 1 (33.3) | 0 | 2 (33.3) | 1 (16.7) | 0 | 0 | 1 (33.3) | 0 | 0 | 0 | 0 | 0 | 1 (33.3) | 0 | 1 (33.3) | 0 | 2 (33.3) | 0 | 4 (10.5) | 0 |
| Infusion-related reaction | 0 | 0 | 0 | 0 | 0 | 0 | 0 | 0 | 0 | 0 | 1 (25.0) | 0 | 1 (33.3) | 0 | 0 | 0 | 0 | 0 | 4 (10.5) | 1 (2.6) |
| Hypomagnesaemia | 0 | 0 | 0 | 0 | 0 | 0 | 0 | 0 | 2 (33.3) | 0 | 0 | 0 | 0 | 0 | 0 | 0 | 1 (16.7) | 0 | 3 (7.9) | 0 |
| Aspartate aminotransferase increased | 0 | 0 | 0 | 0 | 0 | 0 | 0 | 0 | 1 (16.7) | 0 | 1 (25.0) | 1 (25.0) | 0 | 0 | 0 | 0 | 0 | 0 | 2 (5.3) | 1 (2.6) |
| Blood bilirubin increased | 0 | 0 | 0 | 0 | 0 | 0 | 0 | 0 | 1 (16.7) | 0 | 1 (25.0) | 1 (25.0) | 0 | 0 | 0 | 0 | 0 | 0 | 2 (5.3) | 1 (2.6) |
| Hyperuricaemia | 0 | 0 | 0 | 0 | 0 | 0 | 0 | 0 | 1 (16.7) | 0 | 1 (25.0) | 0 | 0 | 0 | 0 | 0 | 1 (16.7) | 0 | 2 (5.3) | 0 |
| Dizziness | 0 | 0 | 1 (16.7) | 0 | 0 | 0 | 0 | 0 | 0 | 0 | 0 | 0 | 0 | 0 | 0 | 0 | 0 | 0 | 2 (5.3) | 0 |

TEAE is defined as any event that occurs on or after the first dose of study drug administration or any pre-existing event which worsened in severity after dosing. Safety Population - Dose Escalation = Patients with at least one dose. Dose groups represent Patients' initial dose. Patients with multiple unique events are counted once per each unique preferred term and System Organ Class. Coding used MedDRA version 23.0. Any event not graded will have a default value of grade 3.

Based on this data, the RP2D of 11.8 mg/kg twice weekly was selected based on the observation that PK exposure levels were similar from 8.8 to 15.6 mg/kg with no increased dose-related AEs or toxicities were observed.

**Efficacy**. While efficacy was not a primary readout for the dose escalation trial, VT1021 did demonstrate single-agent activity in multiple patients. Out of 38 patients who received at least one dose of VT1021 in the escalation phase, 28 patients were considered evaluable based on the criteria of completing at least one cycle of treatment with tumor imaging during cycle 2. One patient with metastatic thymoma (Stage 4) achieved confirmed partial response (PR) and remained on treatment for 504 days. Eleven patients had stable disease (SD) in 9 different solid tumor indications, resulting in a disease control rate (DCR) of 42.9% (Table 5).

To better understand the biological activity of VT1021, pre-study biopsies from 25 evaluable patients were analyzed by immunohistochemistry (IHC). Specifically, the expression levels of CD36 and CD47, the two major cell surface receptors for TSP-1, were assayed. The intensity of each marker was analyzed by Image J/Fiji. Biopsies were scored as being either low, medium or high (representative images are shown in Fig. 3). The scores were measured by both the percentage of cells with positive staining of the biomarkers, and by the level of intensity of staining signal. Moreover, patients were further classified as being dual high for both CD36 and CD47, not dual high, or unknown. The percent change in target lesion from baseline (based on the length of the long axis), correlation to dual high CD36 and CD47 status and duration of exposure to VT1021 for the evaluable patients are shown in Fig. 4. Nine of 25 patients with available biopsies were scored as dual high for CD36 and CD47 (36%) (Fig. 4a). Overall, for all evaluable patients the median duration on treatment was 53 days. Out of the 9 patients with dual high CD36 and CD47 expression, 8 achieved SD (89%) with a mean treatment duration of 148 days (Fig. 4b).

**Biomarker analyses**. Since the mechanism of action (MOA) of VT1021 is mediated by the induction of TSP-1 in MDSCs[4,7], we sought to analyze the expression of TSP-1 in pre- and on-study biopsies. The rationale was that induction of TSP-1 expression would functionally reprogram the TME in patient tumors. Although not required by the study protocol, paired pre- and on-study tumor tissue samples were voluntarily obtained from 7 patients and were analyzed by IHC for expression of CD36, CD47 and TSP-1. The seven patients who provided paired biopsy samples were the following: pancreatic cancer at 5.1 mg/kg, prostate cancer at 5.1 mg/kg, uterine carcinosarcoma at 5.1 mg/kg, kidney cancer at 8.8 mg/kg, appendiceal carcinoma at 11.8 mg/kg, and two ovarian cancer at 15.6 mg/kg. VT1021 induced expression of TSP-1 in the TME in all on-study biopsies analyzed, with one representative image shown in Fig. 5. Significantly, analysis of CD36 and CD47 expression revealed no change in pre- vs on-study biopsies (Fig. 5).

Additionally, we assessed the composition of the TME to determine whether VT1021 was able to reprogram the recruited immune and inflammatory cells. Tumor tissue samples from 4 patients were analyzed for quantitative and qualitative changes in MDSCs, T cells and macrophages after VT1021 treatment. The four patients whose paired biopsy samples were used for quantitative biomarker analysis were the following: uterine carcinosarcoma at 5.1 mg/kg, kidney cancer at 8.8 mg/kg, and two ovarian cancer at 15.6 mg/kg. Analysis of the 4 pairs of biopsies revealed that TSP-1 expression was induced in MDSCs in on-study biopsies compared to pre-study (Fig. 5). Moreover, 3

**Table 4 PK parameters by dose cohort.**

| | 0.5 mg/kg $n=3$ | 1.0 mg/kg $n=6$ | 2.0 mg/kg $n=4$ | 3.3 mg/kg $n=3$ | 5.1 mg/kg $n=6$ | 6.6 mg/kg $n=4$ | 8.8 mg/kg $n=3$ | 11.8 mg/kg $n=3$ | 15.6 mg/kg $n=6$ |
|---|---|---|---|---|---|---|---|---|---|
| **Cycle 1 Day 1** | | | | | | | | | |
| $n$ | 3 | 5 | 4 | 3 | 6 | 4 | 3 | 3 | 4 |
| $C_{max}$ (ng/mL) | 825 (199) | 2790 (940) | 5220 (1500) | 6960 (5750) | 16293 (3911) | 15800 (4557) | 29100 (7440) | 26033 (13059) | 43500 (12700) |
| $n$ | 3 | 5 | 4 | 3 | 6 | 4 | 3 | 3 | 4 |
| $t_{max}$ (hr) | 0.5 (0) | 0.5 (0) | 0.5 (0) | 0.5 (0) | 0.5 (0) | 0.5 (0) | 0.5 (0) | 0.5 (0) | 0.5 (0) |
| $n$ | 3 | 5 | 4 | 3 | 6 | 4 | 3 | 3 | 4 |
| $AUC_{0-6}$ (hr*ng/mL) | 1230 (251) | 4480 (1470) | 7300 (1930) | 10900 (8380) | 26267 (9872) | 30625 (16504) | 51400 (19800) | 45733 (28033) | 69500 (20100) |
| $n$ | 3 | 5 | 4 | 3 | 6 | 3 | 3 | 3 | 4 |
| $t_{1/2}$ (hr) | 1.14 (0.233) | 1.61 (0.729) | 1.27 (0.254) | 1.69 (1.05) | 1.28 (0.264) | 1.14 (0.061) | 1.52 (0.427) | 1.21 (0.273) | 1.28 (0.312) |
| **Cycle 1 Day 4** | | | | | | | | | |
| $n$ | 3 | 5 | 4 | 3 | 6 | 4 | 3 | 3 | 4 |
| $C_{max}$ (ng/mL) | 1320 (331) | 3610 (1190) | 6590 (2320) | 8170 (6940) | 20267 (7507) | 19175 (2690) | 24400 (8880) | 25467 (12987) | 48200 (6760) |
| $n$ | 3 | 5 | 4 | 3 | 6 | 4 | 3 | 3 | 4 |
| $t_{max}$ (hr) | 0.5 (0) | | | | | | | | |
| $n$ | 3 | 5 | 4 | 3 | 6 | 4 | 3 | 3 | 4 |
| $AUC_{0-last}$ (hr*ng/mL) | 1810 (465) | 5860 (1870) | 9160 (3130) | 13000 (11200) | 32450 (17605) | 35850 (14061) | 40800 (15200) | 44200 (26008) | 73100 (10800) |
| $n$ | 3 | 5 | 4 | 2 | 6 | 3 | 3 | 3 | 4 |
| $t_{1/2}$ (hr) | 1.18 (0.115) | 1.94 (0.984) | 1.22 (0.178) | 2.14 (1.35) | 1.35 (0.470) | 1.08 (0.051) | 1.26 (0.229) | 1.21 (0.171) | 1.22 (0.229) |
| **Cycle 2 Day 50** | | | | | | | | | |
| $n$ | 3 | 2 | 1 | 3 | 2 | 2 | 3 | 2 | 2 |
| $C_{max}$ (ng/mL) | 966 (225) | 3820 (453) | 6710 | 11400 (1970) | 15700 (849) | 17900 (9260) | 34200 (1120) | 23650 (16617) | 39500 (22600) |
| $n$ | 3 | 2 | 1 | 3 | 2 | 2 | 3 | 2 | 2 |
| $t_{max}$ (hr) | 0.5 (0) | 0.5 (0) | 0.5 | 0.5 (0) | 0.5 (0) | 0.5 (0) | 0.5 (0) | 0.5 (0) | 0.5 (0) |
| $n$ | 3 | 2 | 1 | 3 | 2 | 2 | 3 | 2 | 2 |
| $AUC_{0-6}$ (hr*ng/mL) | 1360 (265) | 5360 (530) | 8790 | 19500 (7100) | 24300 (566) | 29700 (15300) | 53200 (6470) | 38100 (31537) | 55500 (31100) |
| $n$ | 3 | 2 | 1 | 3 | 2 | 2 | 3 | 2 | 2 |
| $t_{1/2}$ (hr) | 1.21 (0.293) | 1.06 (0.051) | 1.02 | 1.99 (1.14) | 1.13 (0.206) | 1.18 (0.153) | 1.21 (0.228) | 1.05 (0.255) | 1.06 (0.131) |

Values shown are mean and (standard deviation).

$AUC_{0-6}$ area under the curve (AUC) from start of infusion to 6 h post end of-infusion, $C_{max}$ maximum concentration, $n$ number of concentration vs time profiles included in the summary statistics, $t_{1/2}$ terminal phase half-life, $t_{max}$ time-of maximum concentration.

---

**Table 5 Best overall response (assessed by investigators according to RECIST v1.1).**

| | All patients $N=28$ |
|---|---|
| Best overall response, $n$ (%) | 1 (3.6) |
| Complete response | 0 |
| Partial response (confirmed) | 1 (3.6) |
| Stable disease | 11 (39.3) |
| Progressive disease | 16 (57.1) |
| Disease control rate, % (90% CI) | 42.9 (27–60) |

The 90% Confidence Interval (CI) was calculated using the exact (Clopper-Pearson) interval. Best overall response is calculated by including "complete responses" and "partial responses". Disease control rate is calculated by including "complete responses", "partial responses" and "stable diseases".
RECIST Response Evaluation Criteria in Solid Tumors.

out of 4 on-study biopsies displayed increased CD8+ CTLs and iNOS+ M1 macrophages with concomitant decreases in FoxP3+ regulatory T cells and CD163 + M2 macrophages. Figure 5 depicts a representative example of changes observed in the TME in a patient with metastatic renal cell carcinoma (RCC) who achieved SD in the 8.8 mg/kg cohort and was on treatment for 105 days. Similar TSP-1 induction and macrophages repolarization results were also found in the MIBI study (Supplementary Information).

## Discussion

We report here the first-in-human experience of VT1021 in patients with advanced solid tumors who were refractory to multiple lines and various classes of systemic therapies. The study rigorously assessed the safety, PK/PD, and clinical activities of this first-in-class agent, which targets the cell surface molecules CD36 and CD47 simultaneously, via induction of TSP-1[15]. Treatment with VT1021 in this population was safe and well tolerated. The major drug-related toxicity was infusion reaction noted at the protocol-defined starting dose which resolved following pre-medications with steroids and/or antihistamines. The incidence of ≥grade 3 AEs suspected to be related to the study drug was very low (7.9%). There were several patients that died on the study however none were attributed to the study drug, determined by the clinical principal investigators. The patient population in the escalation phase was all heavily pre-treated with more than four previous lines of therapy and multiple metastases. The PK parameters were characterized for each dose cohort and

were observed to increase proportionally from 0.5 mg/kg to 8.8 mg/kg while the exposures from 8.8 mg/kg to 15.6 mg/kg were similar. Because the MTD was not reached, the RP2D of 11.8 mg/kg was selected based on PK exposure. The dosing schedule of 11.8 mg/kg twice weekly was further evaluated in tumor type-specific expansion cohorts, namely GBM, ovarian and pancreatic cancer which will be reported when survival data is fully mature. Out of 28 evaluable patients reported in this study, one patient achieved PR and 11 patients had SD with a DCR of 42.9%. Of the SD patients who provided biopsies, 72.7% had dual high expression of both CD36 and CD47. Biomarker analyses in tumor biopsies confirmed the mechanism of action of VT1021 to induce expression of TSP-1 in MDSCs and reprogram the TME from immunologically cold to hot. Taken together these findings support the clinical advancement of VT1021 into phase 1b/ II single agent and/or combinatorial studies.

The clinical activity of VT1021 in the dose escalation portion of this phase 1 study indicates the potential efficacy in select solid tumor indications which typically harbor an immunologically cold TME and for which treatment with drugs such as checkpoint inhibitors have shown very little benefit[16]. One patient with metastatic thymoma, the only patient with this indication in this study, achieved PR after the second cycle of treatment, and was on study for 504 days, while 11 patients with other solid tumors including pseudomyxoma peritonei ($n = 1$), leiomyosarcoma ($n = 1$), appendiceal ($n = 1$), uterine ($n = 1$), pancreatic ($n = 1$), uterine carcinosarcoma ($n = 1$), kidney ($n = 1$), colorectal ($n = 2$), and ovarian ($n = 2$) had SD. Additionally, exploratory analysis suggests that dual high expression of CD36 and CD47 may predict response. Among 9 patients with dual high expression of CD36 and CD47, 8 patients achieved SD.

Exploratory pharmacodynamic studies on paired tumor biopsies (pre-study and on-study) confirmed the mechanism of action of VT1021. The induction of TSP-1 was observed in all the tested biopsy samples. Although the number of available biopsy pairs was low, modulation of the TME from immunologically cold to hot, another hallmark of VT1021 activity, was also observed by augmented levels of active tumor-killing immune cells and lower levels of immunosuppressive cells in a majority of on-study biopsies. Although VT1021 thus far has been shown to be safe and well tolerated in patients with advanced solid tumors, there are several limitations to this study which will be addressed in future clinical trials such as increasing the number of patients in the treatment group, requiring pre-treatment biopsies from all patients prior to enrollment, and optimizing the dose of single-agent VT1021.

VT1021 is the first clinical-stage molecule that functions by stimulating the expression of TSP-1 in the TME. The stimulation of TSP-1 simultaneously targets both CD36 and CD47 harnessing the full anti-tumor activity of TSP-1. Other drugs have attempted to exploit the anti-tumor activity of TSP-1 by utilizing small regions of the protein[17,18] developed to target CD36 and CD47 individually[15]. VT1021, however, induces the production of endogenous, localized, full-length TSP-1 in MDSCs, potentially improving TSP-1-dependent efficacy. Expression of full-length TSP-1 causes tumor reduction by CD36-dependent induction of apoptosis in tumor cells and endothelial cells. TSP-1 also blocks the CD47-SIRPα "do-not-eat-me" macrophage checkpoint to enable phagocytosis of tumor cells[19,20].

The unique ability of VT1021 to target both CD36 and CD47 concurrently underscores the novel, first-in-class status of this molecule. The expansion phase of this study in selected solid tumor cohorts has recently been completed and results from this, as well as exploration of potential predictive and pharmacodynamic biomarkers, will be reported separately once survival data is more mature. VT1021 is currently in a global registration-ready clinical study (AGILE) for both newly diagnosed and recurrent GBM patients. Additional studies have been planned for other solid tumor indications, as single agent, and as part of the combination regimens with standard of care chemotherapies and immune checkpoint inhibitors.

## Data availability

All data supporting the findings of this study are available within the paper and its Supplementary Information. Individual participant data that underlie the results reported in this paper after deidentified will be shared upon request. Study protocol has been provided in Supplementary Information. Source data for the figures are available as Supplementary Data 1 and Supplementary Data 2. Additional data is available from the corresponding author upon reasonable request. Data requests submitted by researchers who provide a methodologically sound proposal will be accepted beginning immediately following publication and ending 5 years following publication.

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

## Author contributions

Patient enrollment and study monitoring: D.M., W.H., A.P., A.B. Development of methodology for biomarker studies: S.W., M.Y.V., J.J.C., J.W. Acquisition of data: S.W.,

M.Y.V., H.P., J.C., M.C. Analysis and interpretation of data: S.W., M.Y.V., J.J.C., R.S.W., J.C., J.M., M.C., J.W. Study supervision: M.C., J.W.

## Competing interests

Vigeo Therapeutics designed and sponsored the clinical study in this article. R.S.W. is a co-founder of, and consultant for, Vigeo Therapeutics, which has licensed technology from Boston Children's Hospital. M.Y.V., J.J.C., S.W., H.P., J.C., J.M., M.C., and J.W. are employees of Vigeo Therapeutics. D.M., W.H., A.P., and A.B. declare no competing interests.
