## [Peer Review File · Communications Medicine]

Reviewers' comments:

Reviewer #1 (Remarks to the Author):

This manuscript reports on a first-in-human phase I trial of VT1021, a microenvironment modulator that induces the expression of TSP-1 in MDSCs and thus targets CD36 and CD47. With no dose limiting toxicities in 38 patients with advanced solid tumors, the authors conclude that VT1021 is safe and well-tolerated. From preliminary anti-tumor activity data, they deduce that VT1021 may have potential to control the disease and should advance to phase Ib/II of clinical development.

The manuscript is well-written addressing the main questions of the dose escalation phase of the trial. The study protocol seems to have been followed adequately and most of the conclusions the authors draw appear valid. However, the authors should be cautious their conclusions on anti-tumour activity as the main objective of dose escalation phase is assessing safety. The authors do very well in reporting both safety and PK data in great detail and by dose level, but planned analyses of ORR and PFS are missing (see below). I have some specific comments that may help to further improve the quality of the manuscript:

1. As this manuscript reports on the dose escalation but not the dose expansion part of the trial, I suggest adding the term "dose escalation" to the title.
2. I suggest toning down the wording in several sections, especially Translational Relevance and Conclusions, since the sample size of a phase I trial is naturally small and no definite conclusions can be drawn. Terms that should be worded more cautiously include "strongly positioned", "excellent safety profile", "dramatic reprogramming", "remarkably clean", "strikingly", and "ideal candidate".
3. Abstract: In Purpose, the objectives are described as "[to assess] safety, tolerability, dose-limiting toxicities (DLT), antitumor activity, and pharmacokinetics" and then in Patients and Methods as "to evaluate the safety, pharmacology, tolerability, and preliminary efficacy". I prefer the second option as safety covers tolerability and DLTs and pharmacology includes both PK and PD. I would suggest moving option 2 to Purpose and omitting it in Patients and Methods. Preliminary efficacy would then read better as preliminary anti-tumor activity given the context of a phase I trial.
4. Abstract: In Results, what do the authors mean with "statistically indistinguishable"? It seems no statistical test has been performed to support this. I would suggest rephrasing to "similar". The same applies to two other occurrences of "indistinguishable" in the remainder of the manuscript.
5. Abstract: "Among 8 patients whose tumors expressed high levels of both CD36 and CD47, the DCR was 72.7%". This seems to be the wrong percentage. $72.7\% (=8/11)$ is the proportion of patients with high CD36/CD47 among those with SD.
6. Abstract: I would suggest not speaking of an optimal biological dose, a) because it would usually involve jointly considering toxicity and efficacy and b) as it was not planned in the protocol to establish it. I would restrict the wording to MTD and RP2D. This also applies to another occurrence of optimal biological dose.
7. Methods: CTCAE for safety assessment is mentioned in the Abstract but not in Methods.
8. Methods: I struggle to understand the modification of the 3+3 design and think a figure might help to simplify. "Each dose level would enroll at least one patient. If no dose-limiting toxicity (DLT) was observed, the next patient would be enrolled at the next higher dose level." To me this means that without DLTs in the trial, each dose level should have included only one patient. Especially as there were no DLTs observed, it is important to know what was planned and how the patient numbers for the dose levels became so different. To add to my confusion, the legend for Figure 1 calls the design the "traditional 3+3" design.
9. Methods: There are no statistical methods described. Although no formal testing is performed, this could include the definition of ORR/DCR/PFS, the Clopper-Pearson confidence interval and the analysis populations. For analysis populations, the safety population is currently defined in the legends of Table

2 & 3 and the response evaluable population in Results. The DLT evaluable does not seem to appear in the manuscript but in the protocol (as "evaluable population"). A flow chart of analysis populations would be helpful to see which patients contributed to what analyses.

10. Results: In the text and in Table 1, the percentages for primary tumor type do not seem to be correct. For example, for ovarian cancer, 8/38 should be 21% instead of 22%. The other variables in Table 1 seem to be correct. I did not check the dose level specific percentages.

11. Results: I would suggest moving "The dose escalation consisted of the administration of VT1021 intravenously twice weekly at doses of 0.5, 1, 2, 3.3, 5.1, 6.6, 8.8, 11.8 or 15.6 mg/kg" to Methods, potentially with details on criteria for intra-patient dose escalation.

12. Results: ORR and PFS results are not presented although they are secondary endpoints similar to DCR. Instead duration on treatment is shown, which does not seem one of the originally planned endpoints.

13. Results: It would be interesting to know more about the 7 and 4 patients, respectively, that contributed to the biomarker analyses. As a minimum, dose level and best response would help interpreting the results. This could potentially be covered by the flow chart of analysis populations mentioned above.

14. Discussion: It is speculative to assume a high likelihood of being dually high CD36/CD47 for the patient with PR as there are no data to support this.

15. Discussion: The part on limitations is relatively brief, and the raised issues seem to apply to all phase I trials, not this trial in particular.

16. Conclusions: Read very similar to Translational Relevance, Abstract, and parts of the Discussion.

17. Table 2 & 3: "Patients with multiple unique events are counted once per each unique preferred term and System Organ Class." I assume the worst grade experienced is then recorded. If so, the wording should be updated.

18. Table 5: Please add ORR (and potentially PFS). Abbreviation CI is not explained. The first footnote should read "The 90% CI was calculated using the exact Clopper-Pearson method."

19. Figure 1: In my opinion, the figure does not add a lot of information beyond the patient numbers per dose level that can be taken from the tables. I would suggest updating this figure including decision points. Questions this figure should answer: How many patients were there per dose level (it currently does answer that question)? When were more than 3 patients recruited to a dose level? Why were they recruited to that dose level although there were 3 patients at that dose level already? When did dose escalation happen (e.g., when got the dose escalated from 5.1 to 6.6, after 3 or after 6 patients)?

20. Figure 4: Is the order of patients the same for panels A and B? Maybe they could be combined into one figure with two y axes. Unless I missed it, the dually high patient with PD and the greatest tumour lesion growth is not discussed anywhere in the text. It would be most interesting to see what the authors think why this patient fared so poorly. If A and B are paired, the patient had a short exposure time; did the patient come off treatment for PD or for other reasons?

21. Figure 5: As both panels A and C show point estimates, data should be presented as a point with error bars rather than as dynamite plots. The figure legend should describe what the error bars show (e.g., standard deviation, standard error of the mean, confidence interval), and p-values should also be included in the figure legend rather than in the plot itself. Letters to indicate the panels have gone missing (also in Fig. 2).

22. Figure 6: Legend for panel A reads "Bar graph depicting M1:M2 macrophage ratio in pre- vs on-study biopsy of XXX".

Reviewer #2 (Remarks to the Author):

I would like to commend the authors on this phase I study of a novel first-in-class agent targeting the tumor microenvironment. VT1021 is a synthetic cyclic pentapeptide derived from prosaposin that induces the expression of thrombospondin-1 MDSCs recruited to the tumor microenvironment. The aim is to make cold tumors hot. This first-in-human study's primary aim was to assess the safety, tolerability, DLT, antitumor activity, and pharmacokinetics of VT1021, as a single agent in advanced solid tumors.

38 pts were treated with an ITT objective response rate of 1.7%.
Among the 28 evaluable patients, there was a single PR (3.6%) in a patient with thymoma.

Comment 1 - a) what was the duration of response for this patient?
b) The time on treatment is described but typically the duration of response should be reported. Please clarify. Was this a confirmed partial response per RECIST?
c) Similarly for the 11 pts with stable disease - were these confirmed?
4)(SD) in 11 patients with 9 different types of solid tumors for a disease control rate (DCR) of 42.9%. Please give the actual numbers as well as percentages ie n=X

Comment 2: For patients who have signed a consent form, a pre-study biopsy or archival tumor specimen obtained within 6 months prior to study initiation was collected. – please clarify that all patients enrolled on the study signed an informed consent. Assume this refers to an additional optional consent? Please clarify

Comment 3: 97% ECOG <1 – was there an eligibility violation? Were there any additional eligibility deviations?

Comment 4: Provide additional information regarding the premedication used

Comment 5: Provide more information regarding the hepatic failure and multi-organ failure and sepsis. 7 deaths within 30 days of study treatment requires more information to support the ascertain that this drug had a clean safety profile. "Overall, VT1021 had a remarkably clean safety profile" If 7 of the 38% this would represent 18% of patients who died within 30 days of study drug and additional information regarding the deaths is warranted

Comment 6: a) MIBI and expression of ... Please provide references for this assay
b) and also the cutoffs used to determine high CD36 and CD47

Comment 7: Out of 38 patients who received at least one dose of VT1021 in the escalation phase, 28 patients were considered evaluable based on the criteria of completing at least one cycle of treatment with tumor imaging during cycle 2.

In the ITT report the efficacy too ?2.6% 3.6% for the evaluable pts who received at least one cycle – clarify if this was a confirmed response, same for SD
was imaging done every 2 cycles of 8 weeks - there is a risk of bias in the event of toxicity if the scans were timed with cycles. ie in the event of toxicity the start of cycle 2 could be delayed and bias PFS

Comment 8: duration of response would be helpful – rather than the time on treatment
Median PFS?

Reviewer #3 (Remarks to the Author):

Dear Authors, This is a well conducted and ell written study. The novel agent has activity and appears to have a biomarker that would aid patient selection as further development continues.

Few minor points:

1. there were alot of deaths on study and were not attributed to the study drug- what was felt to be the cause in those patients? was it clearly disease?
2. The authors suggest this is well tolerated, what dose level were the individuals who had on treatment deaths?
3. How is further development planned for this agent- biomarker driven/ not. What was the response rate in those with dual positive or any one marker + versus no positive markers. This maybe useful to readers hoping to do studies.
4. The authors propose checkpoint inhibitor combinations, did they have any information on the patients response to ICI given either before study or after? Was any NGS or other predictive marker study to suggest which pts may benefit from added ICI?

Please see below for our point-by-point response to the reviewers' comments on our manuscript entitled "First-in-human, first-in-class phase I trial of the tumor microenvironment modulator VT1021 in patients with advanced solid tumors".

Reviewer #1 (Remarks to the Author):

This manuscript reports on a first-in-human phase I trial of VT1021, a microenvironment modulator that induces the expression of TSP-1 in MDSCs and thus targets CD36 and CD47. With no dose limiting toxicities in 38 patients with advanced solid tumors, the authors conclude that VT1021 is safe and well-tolerated. From preliminary anti-tumor activity data, they deduce that VT1021 may have potential to control the disease and should advance to phase Ib/II of clinical development.

The manuscript is well-written addressing the main questions of the dose escalation phase of the trial. The study protocol seems to have been followed adequately and most of the conclusions the authors draw appear valid. However, the authors should be cautious their conclusions on anti-tumour activity as the main objective of dose escalation phase is assessing safety. The authors do very well in reporting both safety and PK data in great detail and by dose level, but planned analyses of ORR and PFS are missing (see below). I have some specific comments that may help to further improve the quality of the manuscript:

1. As this manuscript reports on the dose escalation but not the dose expansion part of the trial, I suggest adding the term "dose escalation" to the title.

Response: We agree with the suggestion and have added the term "dose escalation" to the title.

2. I suggest toning down the wording in several sections, especially Translational Relevance and Conclusions, since the sample size of a phase I trial is naturally small and no definite conclusions can be drawn. Terms that should be worded more cautiously include "strongly positioned", "excellent safety profile", "dramatic reprogramming", "remarkably clean", "strikingly", and "ideal candidate".

Response: We agree with the suggestion and have modified the language. Please see page 2, 3, and 8.

3. Abstract: In Purpose, the objectives are described as "[to assess] safety, tolerability, dose-limiting toxicities (DLT), antitumor activity, and pharmacokinetics" and then in Patients and Methods as "to evaluate the safety, pharmacology, tolerability, and preliminary efficacy". I prefer the second option as safety covers tolerability and DLTs and pharmacology includes both PK and PD. I would suggest moving option 2 to Purpose and omitting it in Patients and Methods. Preliminary efficacy would then read better as preliminary anti-tumor activity given the context of a phase I trial.

Response: We agree with the reviewer and have modified the text accordingly. Please see page 2.

4. Abstract: In Results, what do the authors mean with "statistically indistinguishable"? It seems no statistical test has been performed to support this. I would suggest rephrasing to "similar". The same applies to two other occurrences of "indistinguishable" in the remainder of the manuscript.

Response: We agree with the reviewer and have modified the words accordingly. Please see page 2 and 7.

5. Abstract: "Among 8 patients whose tumors expressed high levels of both CD36 and CD47, the DCR was 72.7%". This seems to be the wrong percentage. 72.7% (=8/11) is the proportion of patients with high CD36/CD47 among those with SD.

Response: We thank the reviewer for identifying this mistake. We have corrected the text as stated in results: "Among 9 patients with dual high CD36 and CD47 expression, 8 achieved SD, the DCR was 89%". Please see page 2.

6. Abstract: I would suggest not speaking of an optimal biological dose, a) because it would usually involve jointly considering toxicity and efficacy and b) as it was not planned in the protocol to establish it. I would restrict the wording to MTD and RP2D. This also applies to another occurrence of optimal biological dose.

Response: We agree with the reviewer and have modified the text accordingly. Please see page 3 and page 7.

7. Methods: CTCAE for safety assessment is mentioned in the Abstract but not in Methods.

Response: In the revised version, we have mentioned CTCAE for safety assessment in Methods, please see page 4.

8. Methods: I struggle to understand the modification of the 3+3 design and think a figure might help to simplify. "Each dose level would enroll at least one patient. If no dose-limiting toxicity (DLT) was observed, the next patient would be enrolled at the next higher dose level." To me this means that without DLTs in the trial, each dose level should have included only one patient. Especially as there were no DLTs observed, it is important to know what was planned and how the patient numbers for the dose levels became so different. To add to my confusion, the legend for Figure 1 calls the design the "traditional 3+3" design.

Response: The original design was indeed to enroll one subject per dose level until a DLT is observed. Unfortunately, the first subject dosed at the protocol defined starting dose of 1 mg/kg experienced a grade 3 infusion reaction on the third dose. Even though the infusion reaction was reversible, we decided to dose -deescalate to 0.5 mg/kg, and enrolled 3 subjects at this dose level. Subsequently, the protocol was amended to require premedication of steroids and/or antihistamines (at the PI's discretion) prior to the first infusion, and 3 subjects were enrolled for all following dose cohorts.

9. Methods: There are no statistical methods described. Although no formal testing is performed, this could include the definition of ORR/DCR/PFS, the Clopper-Pearson confidence interval and the analysis populations. For analysis populations, the safety population is currently defined in the legends of Table 2 & 3 and the response evaluable population in Results. The DLT evaluable does not seem to appear in the manuscript but in the protocol (as "evaluable population"). A flow chart of analysis populations would be helpful to see which patients contributed to what analyses.

Response: We thank the reviewer for the suggestions.

Regarding statistical analysis, we have added a statistical analysis section to the Methods (page 5). In terms of end points, we used duration of treatment for the escalation phase instead of PFS. The ORR was based on the data that only one patient achieved a PR. PFS is not a meaningful statistic in a dose escalation study as patients received different dose levels, most of which were below the ultimately determined RP2D.

Regarding DLT, we have described the DLT in the results section of DLTs, PK and RP2D, see page 7.

Regarding “analysis populations”, we have clarified the different populations used for safety assessment and for clinical response assessment, on page 5.

10. Results: In the text and in Table 1, the percentages for primary tumor type do not seem to be correct. For example, for ovarian cancer, 8/38 should be 21% instead of 22%. The other variables in Table 1 seem to be correct. I did not check the dose level specific percentages.

Response: We thank the reviewer for identifying this mistake, we have checked the calculation and corrected the wrong numbers. Please see page 6 and the revised Table 1 (page 13).

11. Results: I would suggest moving "The dose escalation consisted of the administration of VT1021 intravenously twice weekly at doses of 0.5, 1, 2, 3.3, 5.1, 6.6, 8.8, 11.8 or 15.6 mg/kg" to Methods, potentially with details on criteria for intra-patient dose escalation.

Response: Thanks for the suggestion, we have moved the sentence to Methods. Please see page 4.

12. Results: ORR and PFS results are not presented although they are secondary endpoints similar to DCR. Instead duration on treatment is shown, which does not seem one of the originally planned endpoints.

Response: ORR and PFS are the endpoints for expansion cohorts therefore not reported here for the escalation subjects.

13. Results: It would be interesting to know more about the 7 and 4 patients, respectively, that contributed to the biomarker analyses. As a minimum, dose level and best response would help interpreting the results. This could potentially be covered by the flow chart of analysis populations mentioned above.

Response: The seven subjects who provided paired biopsy samples were the following: pancreatic cancer at 5.1 mg/kg, prostate cancer at 5.1 mg/kg, carcinosarcoma at 5.1mg/kg, kidney cancer at 8.8 mg/kg, appendiceal carcinoma at 11.8 mg/kg, and two ovarian cancer at 15.6 mg/kg. The four subjects whose paired biopsy samples were used for quantitative biomarker analysis were the following: carcinosarcoma at 5.1mg/kg, kidney cancer at 8.8 mg/kg, and two ovarian cancer at 15.6 mg/kg. The above information has been incorporated into the manuscript (page 8).

Response: Per protocol, enrolled subjects were not required to provide on study biopsy samples. The paired biopsy samples used in CD36 and CD47 expression evaluation, as well as in biomarker analysis, were not selected based on dose levels and clinical responses, but rather based on subjects' willingness to voluntarily participate in "in-treatment" biopsy procedure and provide samples to Vigeo for biomarker analysis. Therefore, the very small sample size does not allow us to perform correlation analysis between dose levels and best responses.

14. Discussion: It is speculative to assume a high likelihood of being dually high CD36/CD47 for the patient with PR as there are no data to support this.

Response: We appreciate the point raised by the reviewer; we made this assumption based on the high (%) incidence of CD36/CD47 dual high staining in a TMA of thymoma patients. To be scientifically accurate, we have removed this statement.

15. Discussion: The part on limitations is relatively brief, and the raised issues seem to apply to all phase I trials, not this trial in particular.

Response: As this was a relatively standard Phase 1 trial with no stratification or enrichment, we feel the limitations we described were the only ones relevant.

16. Conclusions: Read very similar to Translational Relevance, Abstract, and parts of the Discussion.

Response: We have reorganized the translational relevance (plain language summary), abstract, and discussion; and reduced the redundant phrases.

17. Table 2 & 3: "Patients with multiple unique events are counted once per each unique preferred term and System Organ Class." I assume the worst grade experienced is then recorded. If so, the wording should be updated.

Response: Every unique event is recorded and counted once, not just the most severe or worst grade event is recorded.

18. Table 5: Please add ORR (and potentially PFS). Abbreviation CI is not explained. The first footnote should read "The 90% CI was calculated using the exact Clopper-Pearson method."

Response: We have added ORR and included the full term for CI (confidence interval).

19. Figure 1: In my opinion, the figure does not add a lot of information beyond the patient numbers per dose level that can be taken from the tables. I would suggest updating this figure including decision points. Questions this figure should answer: How many patients were there per dose level (it currently does answer that question)? When were more than 3 patients recruited to a dose level? Why were they recruited to that dose level although there were 3 patients at that dose level already? When did dose escalation happen (e.g., when got the dose escalated from 5.1 to 6.6, after 3 or after 6 patients)?

Response: Figure 1 provides how many patients were enrolled per dose level; decision points have been described in results section of the revised version (page 4).

20. Figure 4: Is the order of patients the same for panels A and B? Maybe they could be combined into one figure with two y axes. Unless I missed it, the dually high patient with PD and the greatest tumour lesion growth is not discussed anywhere in the text. It would be most interesting to see what the authors think why this patient fared so poorly. If A and B are paired, the patient had a short exposure time; did the patient come off treatment for PD or for other reasons?

Response: The order of both graphs is the same. The patient mentioned by the reviewer was an ovarian cancer patient, treated at 15.6 mg/kg. She had 5 target lesions when entered the study, and the sum of all lesions increased by 72% at the end of cycle 2 scan. It was determined that the patient had progressive disease and the treatment was terminated. There are myriad factors (genetic, epigenetic, prior treatments, among others) that could explain the reason the patient was a PD, even with dual high expression of CD36 and CD47. Being an n of 1 for this trial it is virtually impossible to determine that reason with sufficient certainty to warrant its discussion in this manuscript.

We agree with the reviewer that this is an extremely important matter, we are conducting Phase 2/3 trials and will perform a more detailed analysis in the future with the goal of identifying additional criteria that support or inhibit response to VT1021.

21. Figure 5: As both panels A and C show point estimates, data should be presented as a point with error bars rather than as dynamite plots. The figure legend should describe what the error bars show (e.g., standard deviation, standard error of the mean, confidence interval), and p-values should also be included in the figure legend rather than in the plot itself. Letters to indicate the panels have gone missing (also in Fig. 2).

Response: We agree with the reviewer and change the graph type to a point with error bars. We also include the description of error bars and p value in the figure legend. The missing letters to indicate the panels are put in.

22. Figure 6: Legend for panel A reads "Bar graph depicting M1:M2 macrophage ratio in pre- vs on-study biopsy of XXX".

Response: We have revised the legend as "Graph of a point with error bars depicting M1:M2 macrophage ratio in pre- vs on-study biopsy of a subject with ovarian cancer". Please see page 25.

Reviewer #2 (Remarks to the Author):

I would like to commend the authors on this phase I study of a novel first-in-class agent targeting the tumor microenvironment. VT1021 is a synthetic cyclic pentapeptide derived from prosaposin that induces the expression of thrombospondin-1 MDSCs recruited to the tumor microenvironment. The aim is to make cold tumors hot. This first-in-human study's primary aim was to assess the safety, tolerability, DLT, antitumor activity, and pharmacokinetics of VT1021, as a single agent in advanced solid tumors.

38 pts were treated with an ITT objective response rate of 1.7%.

Among the 28 evaluable patients, there was a single PR (3.6%) in a patient with thymoma.

Comment 1

a) what was the duration of response for this patient?

Response: This patient was 504 days on study. We have added this information in the manuscript, please see page 7.

b) The time on treatment is described but typically the duration of response should be reported. Please clarify. Was this a confirmed partial response per RECIST?

Response: Yes, the PR was confirmed, and the duration of response is approximately 462 days.

c) Similarly for the 11 pts with stable disease - were these confirmed?

Response: Yes, these were confirmed.

4)(SD) in 11 patients with 9 different types of solid tumors for a disease control rate (DCR) of 42.9%. Please give the actual numbers as well as percentages ie n=X

Response: Indications for subjects with SD have been added on page 9.

Comment 2: For patients who have signed a consent form, a pre-study biopsy or archival tumor specimen obtained within 6 months prior to study initiation was collected. – please clarify that all patients enrolled on the study signed an informed consent. Assume this refers to an additional optional consent? Please clarify

Response: All patients signed an informed consent form to provide a pre-study biopsy. However, there were circumstances that prevented sample collection, such as loss of primary tumor sections from the institution where surgeries were performed, poor quality of sections and slides, etc.

Comment 3: 97% ECOG <1 – was there an eligibility violation? Were there any additional eligibility deviations?

Response: For ESCALTION ECOG was set to be ≤ 2 ; for expansion we modified to ≤ 1 , the text was updated that all patients were ≤ 2

Comment 4: Provide additional information regarding the premedication used

Response: We have added a paragraph in the methods to include additional information regarding the premedication (see page 5)

Comment 5: Provide more information regarding the hepatic failure and multi-organ failure and sepsis. 7 deaths within 30 days of study treatment requires more information to support the ascertain that this drug had a clean safety profile. "Overall, VT1021 had a remarkably clean safety profile" If 7 of the 38% this would represent 18% of patients who died within 30 days of study drug and additional information regarding the deaths is warranted

Response: There might be a wrong interpretation of the time of death of the 4 patients: 3 died within treatment period, and 4 died within 30 days POST treatment termination. NONE of the death is determined to be caused or related to VT1021 treatment. The causes of deaths have been added to the revised draft on page 6.

Comment 6: a) MIBI and expression of ... Please provide references for this assay b) and also the cutoffs used to determine high CD36 and CD47

Response: a) We have added the reference for MIBI assay (page 5, reference #13)

Response: b) Please see revised text on page 8 regarding the cutoffs used to determine high CD36 and CD47.

Comment 7: Out of 38 patients who received at least one dose of VT1021 in the escalation phase, 28 patients were considered evaluable based on the criteria of completing at least one cycle of treatment with tumor imaging during cycle 2.

In the ITT report the efficacy too ?2.6% 3.6% for the evaluable pts who received at least one cycle – clarify if this was a confirmed response, same for SD
was imaging done every 2 cycles of 8 weeks - there is a risk of bias in the event of toxicity if the scans were timed with cycles. ie in the event of toxicity the start of cycle 2 could be delayed and bias PFS

Response: Scans are always timed with cycles in clinical trials unless there is a reason to scan early. The scan results and toxicity of the drug are independent variables. Scans are used to assess efficacy and response. Toxicity is evaluated throughout the course of the study and not only at the end of cycles.

Response: Patients may go off study due to severe adverse events (toxicity) or due to disease progression as determined by imaging. Neither parameter is affected by the other.

Comment 8: duration of response would be helpful – rather than the time on treatment
Median PFS?

Response: Duration of response is not a realistic measurement in our case as there is only one PR, and most of the subjects were not treated at the RP2D.

Reviewer #3 (Remarks to the Author):

Dear Authors,

This is a well conducted and well written study. The novel agent has activity and appears to have a biomarker that would aid patient selection as further development continues.

Few minor points:

1. there were a lot of deaths on study and were not attributed to the study drug- what was felt to be the cause in those patients? was it clearly disease?

Response: We have added additional information in the revised version to address these questions (page 6)

2. The authors suggest this is well tolerated, what dose level were the individuals who had on treatment deaths?

Response: We have added additional information in the revised version to address these questions (page 6)

3. How is further development planned for this agent- biomarker driven/ not. What was the response rate in those with dual positive or any one marker + versus no positive markers. This maybe useful to readers hoping to do studies.

Response: VT1021 is currently in a global registration ready clinical study (AGILE) for both newly diagnosed and recurrent GBM patients. Additional studies have been planned for other solid tumor indications, as single agent, and as part of the combination regimens with SoC and ICI.

4. The authors propose checkpoint inhibitor combinations, did they have any information on the patients response to ICI given either before study or after? Was any NGS or other predictive marker study to suggest which pts may benefit from added ICI?

Response: Preliminary data from various animal model studies demonstrated greater efficacy from the combination of VT1021 and ICI when comparing to ICI alone. No clinical studies have been initiated yet.

Reviewers' comments:

Reviewer #1 (Remarks to the Author):

I thank the authors for addressing my initial comments and do not have any other issues to raise.

Reviewer #2 (Remarks to the Author):

Thank you for the opportunity to review the manuscript. The authors have satisfactorily addressed most of the prior comments.

1) safety and tolerability: "None of the deaths were related to VT1021 treatment."

Would revise this to "None of the deaths were attributed to VT1021 treatment" to mirror the language in the discussion section "none were attributed to the study drug as determined by the clinical principal investigators"

2) "Four subjects died on treatment, and the cause of death were hepatic failure due to disease progression (at 5.1 mg/kg dose level), hepatic failure due to disease progression (at 6.6 mg/kg dose level), tumor hemorrhage (at 15.6 mg/kg dose level) and unrelated to protocol (at 15.6 mg/kg dose level)."

- Please clarify the cause of death for the last subject.

3) "Grade 3 RTEAEs were reported in 3 subjects (7.9%) where there was a single occurrence each of infusion related reaction, anemia, and increased aspartate aminotransferase (AST), blood bilirubin..."

- Is it not possible that the drug could have caused hepatic injury that contributed to the hepatic failure that led to death in 2 subjects? Especially as one of the RTEAEs related to abnormal liver function.

4) pancreatic cancer at 5.1 mg/kg, prostate cancer at 5.1 mg/kg, carcinosarcoma at 5.1mg/kg, kidney cancer

Carcinosarcoma is a histologic subtype. Was this uterine CS or ovarian CS or CS of unknown primary site?

5) The duration of stable disease or PFS is not described.

6)The treatment duration is reported. Were patients permitted to continue on treatment after progression of disease (ie unconfirmed PD)?

7) The prior question regarding whether scans were performed every 8 weeks from baseline timepoint or if they were timed with every 2 cycles was not addressed. If the scans were tied to cycles then any delay in treatment due to toxicity could also potentially result in delay of the scan as the cycle start would be delayed while treatment was held. Please clarify the scan schedule.

Results -

8) 71% of subjects receiving ≥ 3 prior treatment regimens

- Is there any additional information regarding the number of priors in this subgroup? There may be a typo in the text as this figure does not match Table 1.

Reviewer #3 (Remarks to the Author):

The authors have very carefully addressed the comments from all reviewers. For reviewer 3, comments 3 and 4- if they could add those responses into the future plans section so readers can be pointed towards what they plan next, otherwise it reads well.

Reviewers' comments received on Aug 10th, 2023:

Reviewer #1 (Remarks to the Author):

I thank the authors for addressing my initial comments and do not have any other issues to raise.

>> we appreciate the reviewer's time and effort in reviewing our response and are pleased to hear that we have addressed their comments satisfactorily.

Reviewer #2 (Remarks to the Author):

Thank you for the opportunity to review the manuscript. The authors have satisfactorily addressed most of the prior comments.

1) safety and tolerability: "None of the deaths were related to VT1021 treatment."

Would revise this to "None of the deaths were attributed to VT1021 treatment" to mirror the language in the discussion section "none were attributed to the study drug as determined by the clinical principal investigators"

>> we thank the reviewer for the suggestion and have revised the language accordingly on page 6.

2) "Four subjects died on treatment, and the cause of death were hepatic failure due to disease progression (at 5.1 mg/kg dose level), hepatic failure due to disease progression (at 6.6 mg/kg dose level), tumor hemorrhage (at 15.6 mg/kg dose level) and unrelated to protocol (at 15.6 mg/kg dose level)."

- Please clarify the cause of death for the last subject.

>> The death of the last subject was described by the PI in the database as "septic shock- unrelated to protocol;" the text has been updated to include septic shock on page 6.

3) "Grade 3 RTEAEs were reported in 3 subjects (7.9%) where there was a single occurrence each of infusion related reaction, anemia, and increased aspartate aminotransferase (AST), blood bilirubin..."

- Is it not possible that the drug could have caused hepatic injury that contributed to the hepatic failure that led to death in 2 subjects? Especially as one of the RTEAEs related to abnormal liver function.

>> One of the 2 subjects with hepatic failure was diagnosed with colorectal cancer and received only 6 doses of VT1021 before hepatic failure was reported. The second patient was diagnosed with Ovarian cancer and only received 2 doses of VT1021 prior to the report of hepatic failure. In the SAE reporting process, the PI noted that neither reported case was related to VT1021.

4) pancreatic cancer at 5.1 mg/kg, prostate cancer at 5.1 mg/kg, carcinosarcoma at 5.1mg/kg, kidney cancer

Carcinosarcoma is a histologic subtype. Was this uterine CS or ovarian CS or CS of unknown primary site?

>> We thank the reviewer for pointing this out and apologize for not including the specific site of the carcinosarcoma; the patient had uterine carcinosarcoma. Corrections have been made on pages 8, 9 and 13.

5) The duration of stable disease or PFS is not described.

>> we appreciate the reviewer's suggestion. Below please find the duration of response for the PR and SD subjects, with an average of 155 days (not including the two subjects of withdrew and lost in follow up). Median PFS of the escalation study is 64 days. Since this study was a first-in-human dose escalation study with nine dose levels and 14 different tumor types, it is quite difficult to correlate efficacy to these end points.

Indication	Best response	Duration of response	
Escalation (Thymus)	PR	463	
Escalation (Ovarian)	SD	61	
Escalation (Leiomomyosarcoma)	SD	65	
Escalation (Adenoid Cystic Carcinoma)	SD	22	patient lost in follow up
Escalation (Pseudomyxoma Peritonei)	SD	229	
Escalation (Kidney)	SD	60	
Escalation (Ovaries)	SD	110	
Escalation (Colorectal)	SD	182	
Escalation (Uterus)	SD	174	
Escalation (Appendix)	SD	7	Patient withdrew at D64
Escalation (Colon)	SD	128	
Escalation (Pancreatic)	SD	74	

6) The treatment duration is reported. Were patients permitted to continue on treatment after progression of disease (ie unconfirmed PD)?

>> In this study, per protocol, treatment was stopped if the PI determined that there was either radiographic progression or clinical progression of the disease. In other words, the patients were not allowed to stay on treatment and did not receive a second "confirmatory scan" to confirm PD.

7) The prior question regarding whether scans were performed every 8 weeks from baseline timepoint or if they were timed with every 2 cycles was not addressed. If the scans were tied to cycles then any delay in treatment due to toxicity could also potentially result in delay of the scan as the cycle start would be delayed while treatment was held. Please clarify the scan schedule.

>> Scans were performed at every other even cycle (pre-treatment, C2, C4, C6, ...) unless an earlier than scheduled scan was ordered to confirm suspected disease progression. The schedule of scans was not tied to toxicity.

Results -

8) 71% of subjects receiving ≥ 3 prior treatment regimens

- Is there any additional information regarding the number of priors in this subgroup? There may be a typo in the text as this figure does not match Table 1.

>> We appreciate the reviewer pointing out the inconsistency and have corrected the text to 78.9% on page 6.

Reviewer #3 (Remarks to the Author):

The authors have very carefully addressed the comments from all reviewers. For reviewer 3, comments 3 and 4- if they could add those responses into the future plans section so readers can be pointed towards what they plan next, otherwise it reads well.

>> we appreciate the reviewer's suggestion and have added the language on page 10.

From 1st round of comments:

3. How is further development planned for this agent- biomarker driven/ not. What was the response rate in those with dual positive or any one marker + versus no positive markers. This maybe useful to readers hoping to do studies.

>> VT1021 is currently in a global registration ready clinical study (AGILE) for both newly diagnosed and recurrent GBM patients. Additional studies have been planned for other solid tumor indications, as single agent, and as part of the combination regimens with SoC and ICI.

4. The authors propose checkpoint inhibitor combinations, did they have any information on the patients response to ICI given either before study or after? Was any NGS or other predictive marker study to suggest which pts may benefit from added ICI?

>> Preliminary data from various animal model studies demonstrated greater efficacy from the combination of VT1021 and ICI when comparing to ICI alone. No clinical studies have been initiated yet.

REVIEWERS' COMMENTS:

Reviewer #2 (Remarks to the Author):

My comments have been addressed